# Multi-scale Diffusion Denoised Smoothing

**Jongheon Jeong**    **Jinwoo Shin**
Korea Advanced Institute of Science and Technology (KAIST)
Daejeon, South Korea
{jongheonj, jinwoos}@kaist.ac.kr

## Abstract

Along with recent diffusion models, randomized smoothing has become one of a few tangible approaches that offers adversarial robustness to models at scale, *e.g.*, those of large pre-trained models. Specifically, one can perform randomized smoothing on any classifier via a simple "denoise-and-classify" pipeline, so-called *denoised smoothing*, given that an accurate denoiser is available – such as diffusion model. In this paper, we present scalable methods to address the current trade-off between certified robustness and accuracy in denoised smoothing. Our key idea is to "selectively" apply smoothing among multiple noise scales, coined *multi-scale smoothing*, which can be efficiently implemented with a single diffusion model. This approach also suggests a new objective to compare the *collective* robustness of multi-scale smoothed classifiers, and questions which representation of diffusion model would maximize the objective. To address this, we propose to further fine-tune diffusion model (a) to perform consistent denoising whenever the original image is recoverable, but (b) to generate rather diverse outputs otherwise. Our experiments show that the proposed multi-scale smoothing scheme, combined with diffusion fine-tuning, not only allows strong certified robustness at high noise scales but also maintains accuracy close to non-smoothed classifiers. Code is available at https://github.com/jh-jeong/smoothing-multiscale.

## 1   Introduction

Arguably, one of the important lessons in modern deep learning is the effectiveness of massive data and model scaling [38, 74], which enabled many breakthroughs in recent years [8, 53, 55, 7]. Even with the largest amount of data available on the web and computational budgets, however, the *worst-case* behaviors of deep learning models are still challenging to regularize. For example, large language models often leak private information in training data [10], and one can completely fool superhuman-level Go agents with few meaningless moves [40]. Such unintentional behaviors can be of critical concerns in deploying deep learning into real-world systems, and the risk has been increasing as the capability of deep learning continues to expand.

In this context, *adversarial robustness* [61, 9] has been a seemingly-close milestone towards reliable deep learning. Specifically, neural networks are even fragile to small, "imperceptible" scales of noise when it comes to the worst-case, and consequently there have been many efforts to obtain neural networks that are robust to such noise [47, 3, 75, 15, 65]. Although it is a reasonable premise that we humans have an inherent mechanism to correct against adversarial examples [21, 26], yet so far the threat still persists in the context of deep learning, *e.g.*, even in recent large vision-language models [48], and is known as hard to avoid without a significant reduction in model performance [66, 64].

*Randomized smoothing* [41, 15], the focus in this paper, is currently one of a few techniques that has been successful in obtaining adversarial robustness from neural networks. Specifically, it constructs a *smoothed classifier* by taking a majority vote from a "base" classifier, *e.g.*, a neural network, over random Gaussian input noise. The technique is notable for its *provable* guarantees on adversarial

37th Conference on Neural Information Processing Systems (NeurIPS 2023).

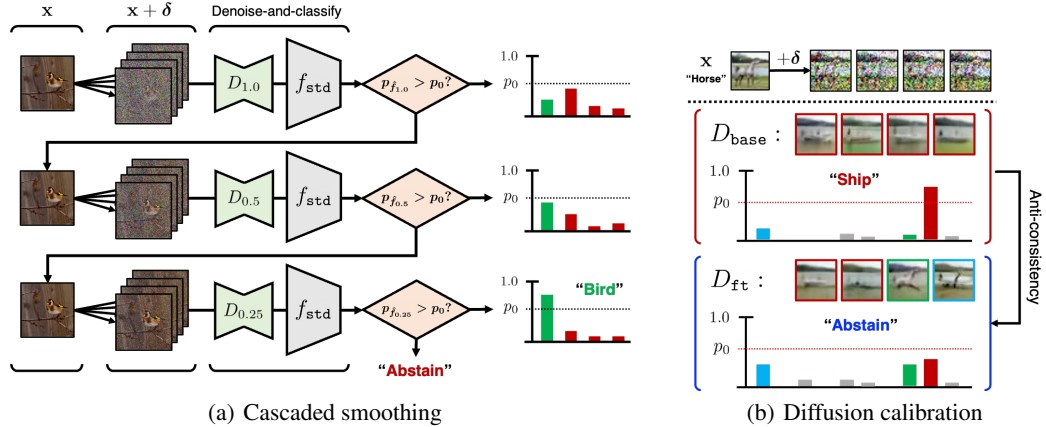

| (a) Cascaded smoothing | (b) Diffusion calibration |

Figure 1: An overview of the proposed approaches, (a) *cascaded randomized smoothing* (Section 3.2) and (b) *diffusion calibration* (Section 3.3) to attain a better trade-off between accuracy and certified robustness in randomized smoothing, upon on the recent *diffusion denoised smoothing* scheme [11].

robustness, *i.e.*, it is a *certified defense* [44], and its scalability to arbitrary model architectures. For example, it was the first certified defense that could offer adversarial robustness on ImageNet [56]. The scalablity of randomized smoothing has further expanded by Salman et al. [58], which observed that randomized smoothing can be applied to any pre-trained classifiers by prepending a denoiser model, dubbed *denoised smoothing*. Combined with the recent *diffusion-based models* [31], denoised smoothing could provide the current state-of-the-arts in $\ell_2$-certified robustness [11, 69, 44].

Despite its desirable properties, randomized smoothing in practice is also at odds with model accuracy [15], similarly to other defenses [47, 76], which makes it burdening to be applied in the real-world. For example, the variance of Gaussian noise, or *smoothing factor*, is currently a crucial hyperparameter to increase certified robustness at the cost of accuracy. The fundamental trade-off between accuracy and adversarial robustness has been well-evidenced in the literature [66, 75, 64], but it has been relatively under-explored whether they may or may not be applicable in the context of randomized smoothing. For example, Tramèr et al. [64] demonstrate that pursuing $\varepsilon$-uniform adversarial robustness in neural networks may increase their vulnerability against *invariance attacks*, *i.e.*, some semantics-altering perturbations possible inside the $\varepsilon$-ball: yet, it is still unclear whether such a vulnerability would be inevitable at non-uniformly robust models such as smoothed classifiers.

In attempt to understand the accuracy-robustness trade-off in randomized smoothing, efforts have been made to increase the certified robustness a given smoothed classifier can provide, *e.g.*, through a better training method [58, 73, 35–37]. For example, Salman et al. [57] have employed adversarial training [47] in the context of randomized smoothing, and Jeong and Shin [35] have proposed to train a classifier with consistency regularization. Motivated by the trade-off among different smoothing factors, some works have alternatively proposed to perform smoothing with *input-dependent* factors [1, 13, 67]: unfortunately, subsequent works [60, 36] have later shown that such schemes do not provide valid robustness certificates, reflecting its brittleness in overcoming the trade-off.

**Contribution.** In this paper, we develop a practical method to overcome the current frontier of accuracy-robustness trade-off in randomized smoothing, particularly upon the architectural benefits that the recent *diffusion denoised smoothing* [11] can offer. Specifically, we first propose to aggregate multiple smoothed classifiers of different smoothing factors to obtain their collective robustness (see Figure 1(a)), which leverages the scale-free nature of diffusion denoised smoothing. In this way, the model can decide which smoothed classifier to use for each input, maintaining the overall accuracy. Next, we fine-tune a diffusion model for randomized smoothing through the "denoise-and-classify" pipeline (see Figure 1(b)), as an efficient alternative of the full fine-tuning of (potentially larger) pre-trained classifiers. Here, we identify the *over-confidence* of diffusion models, *e.g.*, to a specific class given certain backgrounds, as a major challenge towards accurate-yet-robust randomized smoothing, and design a regularization objective to mitigate the issue.

In our experiments, we evaluate our proposed schemes on CIFAR-10 [39] and ImageNet [56], two of standard benchmarks for certified $\ell_2$-robustness, particularly considering practical scenarios of applying diffusion denoised smoothing to large pre-trained models such as CLIP [53]. Overall, the

results consistently highlight that (a) the proposed multi-scale smoothing scheme upon the recent diffusion denoised smoothing [11] can significantly improve the accuracy of smoothed inferences while maintaining their certified robustness at larger radii, and (b) our fine-tuning scheme of diffusion models can additively improve the results - not only in certified robustness but also in accuracy - by simply replacing the denoiser model without any further adaptation. For example, we could improve certifications from a diffusion denoised smoothing based classifier by $30.6\% \rightarrow 72.5\%$ in clean accuracy, while also improving its certified robustness at $\varepsilon = 2.0$ by $12.6\% \rightarrow 14.1\%$. We observe that the collective robustness from our proposed multi-scale smoothing does not always correspond to their individual certified robustness, which has been a major evaluation in the literature, but rather to their "calibration" across models: which opens up a new direction to pursue for a practical use of randomized smoothing.

## 2 Preliminaries

**Adversarial robustness and randomized smoothing.** For a given classifier $f : \mathcal{X} \rightarrow \mathcal{Y}$, where $\mathbf{x} \in \mathcal{X} \subseteq \mathbb{R}^d$ and $y \in \mathcal{Y} := \{1, \cdots, K\}$, *adversarial robustness* refers to the behavior of $f$ in making consistent predictions at the *worst-case* perturbations under semantic-preserving restrictions. Specifically, for samples from a data distribution $(\mathbf{x}, y) \sim p_{\text{data}}(\mathbf{x}, y)$, it requires $f(\mathbf{x} + \boldsymbol{\delta}) = y$ for *every* perturbation $\boldsymbol{\delta}$ that a threat model defines, *e.g.*, an $\ell_2$-ball $\|\boldsymbol{\delta}\|_2 \leq \varepsilon$. One of ways to quantify adversarial robustness is to measure the following *average minimum-distance* of adversarial perturbations [9], *i.e.*, $R(f; p_{\text{data}}) := \mathbb{E}_{(\mathbf{x}, y) \sim p_{\text{data}}} \left[ \min_{f(\mathbf{x}') \neq y} \|\mathbf{x}' - \mathbf{x}\|_2 \right].$

The essential challenge in achieving adversarial robustness stems from that evaluating this (and further optimizing on it) is usually infeasible. *Randomized smoothing* [41, 15] bypasses this difficulty by constructing a new classifier $\hat{f}$ from $f$ instead of letting $f$ to directly model the robustness: specifically, it transforms the base classifier $f$ with a certain *smoothing measure*, where in this paper we focus on the case of Gaussian distributions $\mathcal{N}(0, \sigma^2 \mathbf{I})$:

$$\hat{f}(\mathbf{x}) := \arg\max_{c \in \mathcal{Y}} \mathbb{P}_{\delta \sim \mathcal{N}(0, \sigma^2 \mathbf{I})} \left[ f(\mathbf{x} + \boldsymbol{\delta}) = c \right]. \tag{1}$$

Then, the robustness of $\hat{f}$ at $(\mathbf{x}, y)$, namely $R(\hat{f}; \mathbf{x}, y)$, can be lower-bounded in terms of the *certified radius* $\underline{R}(\hat{f}, \mathbf{x}, y)$, *e.g.*, Cohen et al. [15] showed that the following bound holds, which is tight for $\ell_2$-adversarial threat models:

$$R(\hat{f}; \mathbf{x}, y) \geq \sigma \cdot \Phi^{-1}(p_{\hat{f}}(\mathbf{x}, y)) =: \underline{R}(\hat{f}, \mathbf{x}, y), \quad \text{where} \quad p_{\hat{f}}(\mathbf{x}, y) := \mathbb{P}_{\delta}[f(\mathbf{x} + \boldsymbol{\delta}) = y], \tag{2}$$

provided that $\hat{f}(\mathbf{x}) = y$, otherwise $R(\hat{f}; \mathbf{x}, y) := 0$.[1] Here, we remark that the formula for certified radius is essentially a function of $p_{\hat{f}}$, which is the *accuracy* of $f(\mathbf{x} + \boldsymbol{\delta})$ over $\boldsymbol{\delta}$.

**Denoised smoothing.** Essentially, randomized smoothing requires $f$ to make accurate classification of Gaussian-corrupted inputs. A possible design of $f$ in this regard is to concatenate a Gaussian denoiser, say `denoise`$(\cdot)$, with any standard classifier $f_{\text{std}}$, so-called *denoised smoothing* [58]:

$$f(\mathbf{x} + \boldsymbol{\delta}) := f_{\text{std}}(\texttt{denoise}(\mathbf{x} + \boldsymbol{\delta})). \tag{3}$$

Under this design, an ideal denoiser `denoise`$(\cdot)$ should "accurately" recover $\mathbf{x}$ from $\mathbf{x} + \boldsymbol{\delta}$, *i.e.*, `denoise`$(\mathbf{x} + \boldsymbol{\delta}) \approx \mathbf{x}$ (in terms of their semantics to perform classification) with high probability of $\boldsymbol{\delta} \sim \mathcal{N}(0, \sigma^2 \mathbf{I})$. Denoised smoothing offers a more scalable framework for randomized smoothing, considering that (a) standard classifiers (rather than those specialized to Gaussian noise) are nowadays easier to obtain in the paradigm of large pre-trained models, and (b) the recent developments in diffusion models [31] has supplied denoisers strong enough for the framework. In particular, Lee [42] has firstly explored the connection between diffusion models and randomized smoothing; Carlini et al. [11] has further observed that latest diffusion models combined with a pre-trained classifier provides a state-of-the-art design of randomized smoothing.

**Diffusion models.** In principle, *diffusion models* [59, 31] aims to generate a given data distribution $p_{\text{data}}(\mathbf{x})$ via an iterative denoising process from a Gaussian noise $\hat{\mathbf{x}}_T \sim \mathcal{N}(0, T^2 \mathbf{I})$ for a certain $T > 0$. Specifically, it first assumes the following diffusion process which maps $p_{\text{data}}$ to $\mathcal{N}(0, T^2 \mathbf{I})$: $\mathrm{d}\mathbf{x}_t = \boldsymbol{\mu}(\mathbf{x}_t, t)\mathrm{d}t + \sigma(t)\mathrm{d}\mathbf{w}_t$, where $t \in [0, T]$, and $\mathbf{w}_t$ denotes the standard Brownian motion. Based

---

[1]$\Phi$ denotes the cumulative distribution function of $\mathcal{N}(0, 1^2)$.

on this, diffusion models first train a *score model* $\mathbf{s}_\phi(\mathbf{x}, t) \approx \nabla \log p_t(\mathbf{x})$ via score matching [34], and use the model to solve the probabilistic flow from $\hat{\mathbf{x}}_T \sim \mathcal{N}(0, T^2\mathbf{I})$ to $\mathbf{x}_0$ for sampling. The score estimator $\mathbf{s}_\phi(\mathbf{x}, t)$ is often parametrized by a *denoiser* $D(\mathbf{x}; \sigma(t))$ in practice, *viz.*, $\nabla \log p_t(\mathbf{x}) = (D(\mathbf{x}; \sigma(t)) - \mathbf{x})/\sigma(t)^2$, which establishes its close relationship with denoised smoothing.

## 3  Method

Consider a classification task from $\mathcal{X}$ to $\mathcal{Y}$ where training data $\mathcal{D} = \{(\mathbf{x}_i, y_i)\} \sim p_{\text{data}}(\mathbf{x}, y)$ is available, and let $f : \mathcal{X} \to \mathcal{Y}$ be a classifier. We denote $\hat{f}_\sigma$ to be the smoothed classifier of $f$ with respect to the smoothing factor $\sigma > 0$, as defined by (1). In this work, we aim to better understand how the *accuracy-robustness trade-off* of $\hat{f}_\sigma$ occurs, with a particular consideration of the recent denoised smoothing scheme. Generally speaking, the trade-off implies the following: for a given model, there exists a sample that the model gets "wrong" as it is optimized for (adversarial) robustness on another sample. In this respect, we start by taking a closer look at what it means by a model gets wrong, particularly when it is from randomized smoothing.

### 3.1  Over-smoothing and over-confidence in randomized smoothing

Consider a smoothed classifier $\hat{f}_\sigma$, and suppose there exists a sample $(\mathbf{x}, y)$ where $\hat{f}_\sigma$ makes an error: *i.e.*, $\hat{f}_\sigma(\mathbf{x}) \neq y$. Our intuition here is to separate possible scenarios of $\hat{f}_\sigma(\mathbf{x})$ making an error into two distinct cases, based on the *prediction confidence* of $\hat{f}_\sigma(\mathbf{x})$. Specifically, we define the *confidence* of a smoothed classifier $\hat{f}_\sigma$ at $\mathbf{x}$ based on the definition of randomized smoothing (1) and (2):

$$p_{\hat{f}_\sigma}(\mathbf{x}) := \max_y \ p_{\hat{f}_\sigma}(\mathbf{x}, y) = \max_y \ \mathbb{P}_{\boldsymbol{\delta} \sim \mathcal{N}(0, \sigma^2\mathbf{I})}[f(\mathbf{x} + \boldsymbol{\delta}) = y]. \tag{4}$$

Intuitively, this notion of "smoothed" confidence measures how *consistent* the base classifier $f$ is in classifying $\mathbf{x} + \boldsymbol{\delta}$ over Gaussian noise $\boldsymbol{\delta} \sim \mathcal{N}(0, \sigma^2\mathbf{I})$. In cases when $f$ is modeled by denoised smoothing, achieving high confidence requires the denoiser $D$ to accurately "bounce-back" a given noisy image $\mathbf{x} + \boldsymbol{\delta}$ into one that falls into class $y$ with high probability over $\boldsymbol{\delta}$.

Given a smoothed confidence $p := p_{\hat{f}_\sigma}(\mathbf{x})$, we propose to distinguish two cases of model errors, which are both peculiar to randomized smoothing, by considering a certain threshold $p_0$ on $p$. Namely, we interpret an error of $\hat{f}_\sigma$ at $\mathbf{x}$ either as (a) $p \leq p_0$: the model is *over-smoothing*, or (b) $p > p_0$: the model is having an *over-confidence* on the input:

1. **Over-smoothing ($p \leq p_0$):**  On one hand, it is unavoidable that the mutual information $I(\mathbf{x} + \boldsymbol{\delta}; y)$ between input and its class label absolutely decrease from smoothing with larger variance $\sigma^2$, although using larger $\sigma$ can increase the maximum certifiable radius of $\hat{f}_\sigma$ in practice (2). Here, a "well-calibrated" smoothed classifier should output a prediction close to the uniform distribution across $\mathcal{Y}$, leading to a low prediction confidence. In terms of denoised smoothing, the expectation is clearer: as $\mathbf{x} + \boldsymbol{\delta}$ gets closer to the pure Gaussian noise, the denoiser $D$ should generate more diverse outputs hence in $\mathcal{Y}$ as well. Essentially, this corresponds to an accuracy-robustness trade-off from choosing a specific $\sigma$ for a given (*e.g.*, information-theoretic) capacity of data.

2. **Over-confidence ($p > p_0$):**  On the other hand, it is also possible for a model $\hat{f}_\sigma$ to be incorrect but with a *high confidence*. Compared to the over-smoothing case, this scenario rather signals a "miscalibration" and corresponds to a trade-off from *model biases*: even across smoothed models with a fixed $\sigma$, the balance between accuracy and robustness can be different depending on how each model assigns robustness in its decision boundary per-sample basis. When viewed in terms of denoised smoothing, this occurrence can reveal an implicit bias of the denoiser function $D$, *e.g.*, that of diffusion models. For example, a denoiser might be trained to adapt to some spurious cues in training data, *e.g.*, their backgrounds, as also illustrated in Figure 1(b).

In the subsequent sections, Section 3.2 and 3.3, we introduce two methods to exhibit better accuracy-robustness trade-off in randomized smoothing, each of which focuses on the individual scenarios of over-smoothing and over-confidence, respectively. Specifically, Section 3.2 proposes to use a *cascaded inference* of multiple smoothed classifiers across different smoothing factors to mitigate the limit of using a single smoothing factor. Next, in Section 3.3, we propose to calibrate diffusion models to reduce its over-confidence particularly in denoised smoothing.

### 3.2 Cascaded randomized smoothing

To overcome the trade-off between accuracy and certified robustness from *over-smoothing*, *i.e.*, from choosing a specific $\sigma$, we propose to combine *multiple* smoothed classifiers with different $\sigma$'s. In a nutshell, we design a pipeline of smoothed inferences that each input (possibly with different noise resilience) can adaptively select which model to use for its prediction. Here, the primary challenge is to make it "correct", so that the proposed pipeline does not break the existing statistical guarantees on certified robustness that each smoothed classifier makes.

Specifically, we now assume $K$ distinct smoothing factors, say $0 < \sigma_1 < \cdots < \sigma_K$, and their corresponding smoothed classifiers of $f$, namely $\hat{f}_{\sigma_1}, \cdots, \hat{f}_{\sigma_K}$. For a given input $\mathbf{x}$, our desiderata is (a) to maximize robustness certification at $\mathbf{x}$ based on the smoothed inferences available from the individual models, say $p_{\hat{f}_{\sigma_1}}(\mathbf{x}), \cdots, p_{\hat{f}_{\sigma_K}}(\mathbf{x})$, while (b) minimizing the access to each of the models those require a separate Monte Calro integration in practice. In these respects, we propose a simple "predict-or-abstain" policy, coined *cascaded randomized smoothing*:[2]

$$\texttt{casc}(\mathbf{x}; \{\hat{f}_{\sigma_i}\}_{i=1}^K) := \begin{cases} \hat{f}_{\sigma_K}(\mathbf{x}) & \text{if } p_{\hat{f}_{\sigma_K}}(\mathbf{x}) > p_0, \\ \texttt{casc}(\mathbf{x}; \{\hat{f}_{\sigma_i}\}_{i=1}^{K-1}) & \text{if } p_{\hat{f}_{\sigma_K}}(\mathbf{x}) \le p_0 \text{ and } K > 1, \\ \texttt{ABSTAIN} & \text{otherwise}, \end{cases} \tag{5}$$

where $\texttt{ABSTAIN} \notin \mathcal{Y}$ denotes an artificial class to indicate "undecidable". Intuitively, the pipeline starts from computing $\hat{f}_{\sigma_K}(\mathbf{x})$, the model with highest $\sigma$, but takes its output only if its (smoothed) confidence $p_{\hat{f}_{\sigma_K}}(\mathbf{x})$ exceeds a certain threshold $p_0$:[3] otherwise, it tries a smaller noise scale, say $\sigma_{K-1}$ and so on, applying the same abstention policy of $p_{\hat{f}_\sigma}(\mathbf{x}) \le p_0$. In this way, it can early-stop the computation at higher $\sigma$ if it is confident enough, so it can maintain higher certified robustness, while avoiding unnecessary accesses to other models of smaller $\sigma$.

Next, we ask whether this pipeline can indeed provide a robustness certification: *i.e.*, how much one can ensure $\texttt{casc}(\mathbf{x} + \boldsymbol{\delta}) = \texttt{casc}(\mathbf{x})$ in its neighborhood $\boldsymbol{\delta}$. Theorem 3.1 below shows that one can indeed enjoy the most certified radius from $\hat{f}_{\sigma_k}$ where $\texttt{casc}(\mathbf{x}) =: \hat{y}$ halts, as long as the preceding models are either keep abstaining or output $\hat{y}$ over $\boldsymbol{\delta}$:[4]

**Theorem 3.1.** *Let $\hat{f}_{\sigma_1}, \cdots, \hat{f}_{\sigma_K} : \mathcal{X} \to \mathcal{Y}$ be smoothed classifiers with $0 < \sigma_1 < \cdots < \sigma_K$. Suppose $\texttt{casc}(\mathbf{x}; \{\hat{f}_{\sigma_i}\}_{i=1}^K) =: \hat{y} \in \mathcal{Y}$ halts at $\hat{f}_{\sigma_k}$ for some $k$. Consider any $\underline{p}$ and $\overline{p}_{i,c} \in [0,1]$ that satisfy the following:* **(a)** $\underline{p} \le p_{\hat{f}_{\sigma_k}}(\mathbf{x}, \hat{y})$, *and* **(b)** $\overline{p}_{k',c} \ge p_{\hat{f}_{\sigma_{k'}}}(\mathbf{x}, c)$ *for* $k' > k$ *and* $c \in \mathcal{Y}$*. Then, it holds that* $\texttt{casc}(\mathbf{x} + \boldsymbol{\delta}; \{\hat{f}_{\sigma_i}\}_{i=1}^K) = \hat{y}$ *for any* $\|\boldsymbol{\delta}\| < R$*, where:*

$$R := \min\left\{ \sigma_k \cdot \Phi^{-1}\left(\underline{p}\right), \min_{\substack{y \neq \hat{y} \\ k' > k}} \left\{ \sigma_{k'} \cdot \Phi^{-1}\left(1 - \overline{p}_{k',y}\right) \right\} \right\}. \tag{6}$$

Overall, the proposed multi-scale smoothing scheme (and Theorem 3.1) raise the importance of "abstaining well" in randomized smoothing: if a smoothed classifier can perfectly detect and abstain its potential errors, one could overcome the trade-off between accuracy and certified robustness by joining a more accurate model afterward. The option to abstain in randomized smoothing was originally adopted to make its statistical guarantees correct in practice. Here, we extend this usage to also rule out less-confident predictions for a more conservative decision making. As discussed in Section 3.1, now the *over-confidence* becomes a major challenge in this matter: *e.g.*, such samples can potentially bypass the abstention policy of cascaded smoothing, which motivates our fine-tuning scheme presented in Section 3.3.

**Certification.** We implement our proposed cascaded smoothing to make "statistically consistent" predictions across different noise samples, considering a certain *significance level* $\alpha$ (*e.g.*, $\alpha = 0.001$): in a similar fashion as Cohen et al. [15]. Roughly speaking, for a given input $\mathbf{x}$, it makes predictions only when the $(1 - \alpha)$-confidence interval of $p_{\hat{f}}(\mathbf{x})$ does not overlap with $p_0$ upon $n$ *i.i.d.* noise samples (otherwise it abstains). The more details can be found in Appendix D.2.

---

[2]We also discuss several other possible (and more sophisticated) designs in Appendix E.
[3]In our experiments, we simply use $p_0 = 0.5$ for all $\sigma$'s. See Appendix C.3 for an ablation study with $p_0$.
[4]The proof of Theorem 3.1 is provided in Appendix D.1.

## 3.3 Calibrating diffusion models through smoothing

Next, we move on to the *over-confidence* issue in randomized smoothing: *viz.*, $\hat{f}_\sigma$ often makes errors with a high confidence to wrong classes. We propose to fine-tune a given $\hat{f}_\sigma$ to make rather *diverse* outputs when it misclassifies, *i.e.*, towards a more "calibrated" $\hat{f}_\sigma$. By doing so, we aim to cast the issue of over-confidence as that of *over-smoothing*: which can be easier to handle in practice, *e.g.*, by abstaining. In this work, we particularly focus on fine-tuning only the *denoiser model* $D$ in the context of denoised smoothing, *i.e.*, $f := f_{\tt std} \circ D$ for a standard classifier $f_{\tt std}$: this offers a more scalable approach to improve certified robustness of pre-trained models, given that fine-tuning the entire classifier model in denoised smoothing can be computationally prohibitive in practice.

Specifically, given a base classifier $f := f_{\tt std} \circ D$ and training data $\mathcal{D}$, we aim to fine-tune $D$ to improve the certified robustness of $\hat{f}_\sigma$. To this end, we leverage the *confidence* information of the backbone classifier $f_{\tt std}$ fairly assuming it as an "oracle" - which became somewhat reasonable given the recent off-the-shelf models available - and apply different losses depending on the confidence information per-sample basis. We propose two losses in this matter: (a) *Brier loss* for either correct or under-confident samples, and (b) *anti-consistency loss* for incorrect, over-confident samples.

**Brier loss.** For a given training sample $(\mathbf{x}, y)$, we adopt the Brier (or "squared") loss [6] to regularize the denoiser function $D$ to promote the confidence of $f_{\tt std}(D(\mathbf{x} + \boldsymbol{\delta}))$ towards $y$, which can be beneficial to increase the smoothed confidence $p_{\hat{f}_\sigma}(\mathbf{x})$ that impacts the certified robustness at $\mathbf{x}$. Compared to the cross-entropy (or "log") loss, a more widely-used form in such purpose, we observe that the Brier loss can be favorable in such fine-tuning of $D$ through $f_{\tt std}$, in a sense that the loss is less prone to "over-optimize" the confidence at values closer to 1.

Here, an important detail is that we do not apply the regularization to incorrect-yet-confident samples: *i.e.*, whenever $f_{\tt std}(D(\mathbf{x} + \boldsymbol{\delta})) \neq y$ and $p_{\tt std}(\boldsymbol{\delta}) := \max_c F_{{\tt std},c}(D(\mathbf{x} + \boldsymbol{\delta})) > p_0$, where $F_{\tt std}$ is the soft prediction of $f_{\tt std}$. This corresponds to the case when $D(\mathbf{x} + \boldsymbol{\delta})$ rather outputs a "realistic" off-class sample, which will be handled by the anti-consistency loss we propose. Overall, we have:

$$L_{\tt Brier}(\mathbf{x}, y) := \mathbb{E}_{\boldsymbol{\delta}}[\mathbf{1}[\hat{y}_{\boldsymbol{\delta}} = y \text{ or } p_{\tt std}(\boldsymbol{\delta}) \leq p_0] \cdot \|F_{\tt std}(D(\mathbf{x} + \boldsymbol{\delta})) - \mathbf{e}_y\|^2], \tag{7}$$

where we denote $\hat{y}_{\boldsymbol{\delta}} := f(\mathbf{x} + \boldsymbol{\delta})$ and $\mathbf{e}_y$ is the $y$-th unit vector in $\mathbb{R}^{|\mathcal{Y}|}$.

**Anti-consistency loss.** On the other hand, the anti-consistency loss aims to detect whether the sample is over-confident, and penalizes it accordingly. The challenge here is that identifying over-confidence in a smoothed classifier requires checking for $p_{\hat{f}_\sigma}(\mathbf{x}) > p_0$ (4), which can be infeasible during training. We instead propose a simpler condition to this end, which only takes two independent Gaussian noise, say $\boldsymbol{\delta}_1, \boldsymbol{\delta}_2 \sim \mathcal{N}(0, \sigma^2 \mathbf{I})$. Specifically, we identify $(\mathbf{x}, y)$ as over-confident whenever (a) $\hat{y}_1 := f(\mathbf{x} + \boldsymbol{\delta}_1)$ and $\hat{y}_2 := f(\mathbf{x} + \boldsymbol{\delta}_2)$ match, while (b) they are incorrect, *i.e.*, $\hat{y}_1 \neq y$. Intuitively, such a case signals that the denoiser $D$ is often making a complete flip to the semantics of $\mathbf{x} + \boldsymbol{\delta}$ (see Figure 1(b) for an example), which we aim to penalize. A care should be taken, however, considering the possibility that $D(\mathbf{x} + \boldsymbol{\delta})$ indeed generates an in-distribution sample that falls into a different class: in this case, penalizing it may result in a decreased robustness of that sample. In these respects, our design of anti-consistency loss forces the two samples simply to have different predictions, by keeping at least one prediction as the original, while penalizing the counterpart. Denoting $\mathbf{p}_1 := F_{\tt std}(D(\mathbf{x} + \boldsymbol{\delta}_1))$ and $\mathbf{p}_2 := F_{\tt std}(D(\mathbf{x} + \boldsymbol{\delta}_2))$, we again apply the squared loss on $\mathbf{p}_1$ and $\mathbf{p}_2$ to implement the loss design, as the following:

$$L_{\tt AC}(\mathbf{x}, y) := \mathbf{1}[\hat{y}_1 = \hat{y}_2 \text{ and } \hat{y}_1 \neq y] \cdot (\|\mathbf{p}_1 - {\tt sg}(\mathbf{p}_1)\|^2 + \|\mathbf{p}_2\|^2), \tag{8}$$

where ${\tt sg}(\cdot)$ denotes the stopping gradient operation.

**Overall objective.** Combining the two proposed losses, *i.e.*, the Brier loss and anti-consistency loss, defines a new regularization objective to add upon any pre-training objective for the denoiser $D$:

$$L(D) := L_{\tt Denoiser} + \lambda \cdot (L_{\tt Brier} + \alpha \cdot L_{\tt AC}), \tag{9}$$

where $\lambda, \alpha > 0$ are hyperparameters. Here, $\alpha$ denotes the relative strength of $L_{\tt AC}$ over $L_{\tt Brier}$ in its regularization.[5] Remark that increasing $\alpha$ would give more penalty on over-confident samples, which would lead the model to make more abstentions: therefore, this results in an increased accuracy particularly in cascaded smoothing (Section 3.2).

---

[5] In our experiments, we simply use $\alpha = 1$ unless otherwise specified.

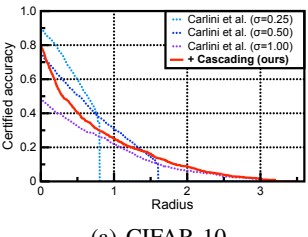

(a) CIFAR-10

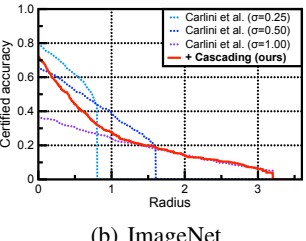

(b) ImageNet

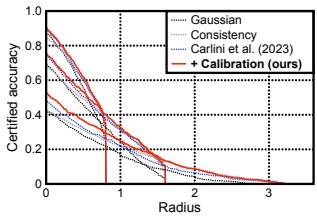

Figure 2: Comparison of certified test accuracy with Carlini et al. [11] on the effectiveness of cascaded smoothing.

Figure 3: Comparison of certified test accuracy on CIFAR-10 at $\sigma \in \{0.25, 0.50, 1.00\}$.

Table 1: Comparison of (a) certified accuracy, (b) empricial accuracy, and (c) average certified radius (ACR) on CIFAR-10. We set our result bold-faced whenever it achieves the best upon baselines.

| CIFAR-10, ViT-B/16@224 | | Empirical accuracy (%) | | | Certified accuracy at $\varepsilon$ (%) | | | | | |
|---|---|---|---|---|---|---|---|---|---|---|
| $\sigma$ | Training | C10 | C10-C | C10.1 | 0.0 | 0.5 | 1.0 | 1.5 | 2.0 | ACR |
| 0.25 | Gaussian [15] | 92.7 | 88.3 | 76.7 | 87.0 | 55.3 | | | | 0.479 |
| | Consistency [35] | 89.5 | 84.7 | 67.6 | 84.9 | 60.7 | | | | 0.517 |
| | Carlini et al. [11] | 92.8 | 88.8 | 67.6 | 89.5 | 59.8 | | | | 0.527 |
| | + Calibration (ours) | **94.5** | **89.4** | 69.2 | **90.3** | **61.9** | | | | **0.535** |
| 0.50 | Gaussian [15] | 87.4 | 81.8 | 71.0 | 69.9 | 45.9 | 24.4 | 6.0 | | 0.543 |
| | Consistency [35] | 80.7 | 75.0 | 59.6 | 67.7 | 49.2 | 31.2 | 13.1 | | 0.617 |
| | Carlini et al. [11] | 88.8 | 84.9 | 64.3 | 75.3 | 50.6 | 31.0 | 14.3 | | 0.646 |
| | + Calibration (ours) | 88.6 | 84.2 | 63.7 | 75.1 | **53.3** | **32.5** | **15.3** | | **0.664** |
| (0.25-0.50) | + Cascading (ours) | **92.5** | **89.1** | **78.5** | **85.1** | 53.3 | 32.9 | 14.9 | | **0.680** |
| 1.00 | Gaussian [15] | 75.6 | 72.4 | 68.3 | 42.2 | 27.9 | 16.9 | 9.0 | 4.0 | 0.426 |
| | Consistency [35] | 65.8 | 61.3 | 53.6 | 43.1 | 31.1 | 22.3 | 14.6 | 8.7 | 0.513 |
| | Carlini et al. [11] | 87.3 | 83.2 | 73.0 | 48.6 | 32.5 | 21.9 | 11.8 | 6.4 | 0.530 |
| | + Calibration (ours) | 83.3 | 79.3 | 60.9 | **52.6** | **36.7** | **24.9** | **16.1** | **8.8** | **0.577** |
| (0.25-1.00) | + Cascading (ours) | **90.1** | **86.9** | **74.5** | **79.6** | **40.5** | **25.0** | **16.2** | **8.8** | **0.645** |

# 4 Experiments

We verify the effectiveness of our proposed schemes, (a) *cascaded smoothing* and (b) *diffusion calibration*, mainly on CIFAR-10 [39] and ImageNet [56]: two standard datasets for an evaluation of certified $\ell_2$-robustness. We provide the detailed experimental setups, *e.g.*, training, datasets, hyperparameters, computes, *etc.*, in Appendix B.

**Baselines.** Our evaluation mainly compares with *diffusion denoised smoothing* [11], the current state-of-the-art methodology in randomized smoothing. We additionally compare with two other training baselines from the literature, by considering models with the same classifier architecture but without the denoising step of Carlini et al. [11]. Specifically, we consider (a) *Gaussian training* [15], which trains a classifier with Gaussian augmentation; and (b) *Consistency* [35], which additionally regularizes the variance of predictions over Gaussian noise in training. We select the baselines assuming practical scenarios where the training cost of the classifier side is crucial: other existing methods for smoothed classifiers often require much more costs, *e.g.*, 8 times over Gaussian [57, 73].

**Setups.** We follow Carlini et al. [11] for the choice of diffusion models: specifically, we use the 50M-parameter $32 \times 32$ diffusion model from Nichol and Dhariwal [52] for CIFAR-10, and the 552M-parameter $256 \times 256$ unconditional model from Dhariwal and Nichol [18] for ImageNet. For the classifier side, we use ViT-B/16 [63] pre-trained via CLIP [53] throughout our experiments. For uses we fine-tune the model on each of CIFAR-10 and ImageNet via FT-CLIP [19], resulting in classifiers that achieve 98.1% and 85.2% in top-1 accuracy on CIFAR-10 and ImageNet, respectively. Following the prior works, we mainly consider $\sigma \in \{0.25, 0.50, 1.00\}$ for smoothing in our experiments.

**Evaluation metrics.** We consider two popular metrics in the literature when evaluating certified robustness of smoothed classifiers: (a) the *approximate certified test accuracy* at $r$: the fraction of the test set which CERTIFY [15] classifies correctly with the radius larger than $r$ without abstaining,

Table 2: Comparison of (a) certified accuracy, (b) empricial accuracy, and (c) average certified radius (ACR) on ImageNet. We set our result bold-faced whenever it achieves the best upon baselines.

| ImageNet, ViT-B/16@224 | | Empirical accuracy (%) | | | Certified accuracy at $\varepsilon$ (%) | | | | | |
|---|---|---|---|---|---|---|---|---|---|---|
| Training | $\sigma$ | IN-1K | IN-R | IN-A | 0.0 | 0.5 | 1.0 | 1.5 | 2.0 | ACR |
| Carlini et al. [11] | 0.25 | 80.9 | 69.3 | 35.3 | 78.8 | 61.4 | | | | 0.517 |
| Carlini et al. [11] | 0.50 | 79.5 | 67.5 | 32.3 | 65.8 | 52.2 | 38.6 | 23.6 | | 0.703 |
| + Cascading (ours) | (0.25-0.50) | **83.5** | **69.6** | **41.3** | **75.0** | **52.6** | **39.0** | 22.8 | | **0.720** |
| + Calibration (ours) | (0.25-0.50) | **83.8** | **69.8** | **41.7** | **76.6** | **54.6** | **39.8** | 23.0 | | **0.743** |
| Carlini et al. [11] | 1.00 | 77.2 | 64.3 | 32.0 | 30.6 | 25.8 | 20.6 | 17.0 | 12.6 | 0.538 |
| + Cascading (ours) | (0.25-1.00) | **82.6** | **69.8** | **40.5** | **69.0** | **42.4** | **26.6** | **19.0** | **14.6** | **0.752** |
| + Calibration (ours) | (0.25-1.00) | **83.2** | **69.5** | **40.8** | **72.5** | **44.0** | **27.5** | **19.9** | 14.1 | **0.775** |

and (b) the *average certified radius* (ACR) [73]: the average of certified radii on the test set $\mathcal{D}_{\texttt{test}}$ while assigning incorrect samples as 0: *viz.*, $\text{ACR} := \frac{1}{|\mathcal{D}_{\texttt{test}}|} \sum_{(\mathbf{x},y) \in \mathcal{D}_{\texttt{test}}} [\text{CR}(f, \sigma, \mathbf{x}) \cdot \mathbb{1}_{\hat{f}(\mathbf{x})=y}]$, where $\text{CR}(\cdot)$ denotes the certified radius that CERTIFY returns. Throughout our experiments, we use $n = 10,000$ noise samples to certify robustness for both CIFAR-10 and ImageNet. We follow [15] for the other hyperparameters to run CERTIFY, namely by $n_0 = 100$, and $\alpha = 0.001$. In addition to certified accuracy, we also compare the *empirical accuracy* of smoothed classifiers. Here, we define empirical accuracy by the fraction of test samples those are either (a) certifiably correct, or (b) abstained but correct in the *clean* classifier: which can be a natural alternative especially at denoised smoothing. For this comparison, we use $n = 100$ to evaluate empirical accuracy.

Unlike the evaluation of Carlini et al. [11], however, we do not compare cascaded smoothing with the *envelop accuracy curve* over multiple smoothed classifiers at $\sigma \in \{0.25, 0.50, 1.00\}$: although the envelope curve can be a succinct proxy to compare methods, it can be somewhat misleading and unfair to compare the curve directly with an individual smoothed classifier. This is because the curve does not really construct a concrete classifier on its own: it additionally assumes that each test sample has prior knowledge on the value of $\sigma \in \{0.25, 0.5, 1.0\}$ to apply, which is itself challenging to infer. This is indeed what our proposal of cascaded smoothing addresses.

### 4.1 Results

**Cascaded smoothing.** Figure 2 visualizes the effect of cascaded smoothing we propose in the plots of certified accuracy, and the detailed results are summarized in Table 1 and 2 as "+ Cascading". Overall, we observe that the certified robustness that cascaded smoothing offers can be highly desirable over the considered single-scale smoothed classifiers. Compared to the single-scale classifiers at highest $\sigma$, *e.g.*, $\sigma = 1.0$ in Table 1 and 2, our cascaded classifiers across $\sigma \in \{0.25, 0.50, 1.00\}$ absolutely improve the certified accuracy at all the range of $\varepsilon$ by incorporating more accurate predictions from classifiers, *e.g.*, of lower $\sigma \in \{0.25, 0.50\}$. On the opposite side, *e.g.*, compared to Carlini et al. [11] at $\sigma = 0.25$, the cascaded classifiers provide competitive certified clean accuracy ($\varepsilon = 0$), *e.g.*, 89.5% *vs.* 85.1% on CIFAR-10, while being capable of offering a wider range of robustness certificates. Those considerations are indeed reflected quantitatively in terms of the improvements in ACRs. The existing gaps in the clean accuracy are in principle due to the errors in higher-$\sigma$ classifiers: *i.e.*, a better calibration, to let them better abstain, could potentially reduce the gaps.

**Diffusion calibration.** Next, we evaluate the effectiveness of our proposed diffusion fine-tuning scheme: "+ Calibration" in Table 1 and 2 report the results on CIFAR-10 and ImageNet, respectively, and Figure 3 plots the CIFAR-10 results for each of $\sigma \in \{0.25, 0.50, 1.00\}$. Overall, on both CIFAR-10 and ImageNet, we observe that the proposed fine-tuning scheme could *uniformly* improve certified accuracy across the range considered, even including the clean accuracy. This confirms that the tuning could essentially improve the accuracy-robustness trade-off rather than simply moving along itself. As provided in Table 3, we remark that simply pursuing only the Brier loss (7) may achieve a better ACR in overall, but with a decreased accuracy: it is the role of the anti-consistency loss (8) to balance between the two, consequently to achieve a better trade-off afterwards.

**Empirical accuracy.** In practical scenarios of adopting smoothed classifiers for inference, it is up to users to decide how to deal with the "abstained" inputs. Here, we consider a possible candidate of simply outputting the *standard* prediction instead for such inputs: in this way, the output could be noted as an "uncertified" prediction, while possibly being more accurate, *e.g.*, for in-distribution

Table 3: Comparison of ACR and certified accuracy of cascaded smooth classifiers on CIFAR-10 over different training and ablations. We use $\sigma \in \{0.25, 0.5, 1.0\}$ for cascaded smoothing. Bold and underline indicate the best and runner-up, respectively. Model reported in Table 1 is marked as grey.

| Cascaded, $\sigma \in \{0.25, 0.5, 1.0\}$ | | Certified accuracy at $\varepsilon$ (%) | | | | | | | |
|---|---|---|---|---|---|---|---|---|---|
| Training | ACR | 0.00 | 0.25 | 0.50 | 0.75 | 1.00 | 1.25 | 1.50 | 1.75 |
| Gaussian [15] | 0.470 | 70.7 | 46.9 | 31.7 | 23.7 | 16.8 | 12.4 | 9.2 | 6.5 |
| Consistency [35] | 0.548 | 57.6 | 42.7 | 32.3 | 26.3 | 22.4 | 18.1 | 14.6 | 11.1 |
| Carlini et al. [11] | 0.579 | 80.5 | 52.8 | 37.4 | 28.0 | 21.7 | 16.8 | 11.8 | 8.5 |
| + Cross-entropy | 0.605 | 77.9 | 52.8 | 39.6 | 29.1 | 22.2 | 18.7 | 13.5 | 9.5 |
| **+ Brier loss** | **0.666** | 77.6 | **55.7** | **41.7** | **32.7** | **26.8** | **21.7** | **16.3** | **11.9** |
|   + Anti-consist. ($\alpha = 1.0$) | 0.645 | 79.6 | 53.5 | 40.5 | 31.2 | 25.0 | 20.1 | 16.2 | 11.4 |
|   + Anti-consist. ($\alpha = 2.0$) | 0.625 | 80.1 | 54.6 | 39.2 | 30.6 | 24.0 | 18.5 | 14.5 | 9.9 |
| **+ Anti-consist.** ($\alpha = 1.0$) | 0.566 | **83.9** | 55.1 | 37.8 | 25.8 | 19.8 | 14.5 | 11.5 | 8.1 |

Table 4: Comparison of ACR and certified accuracy (%) on CIFAR-10 across two fine-tuning schemes: *diffusion* fine-tuning ("Diff."; ours) and *classifier* fine-tuning ("Class.") [11]. We use $\sigma = 0.5$ for this experiment.

| Fine-tuning | | | Certified accuracy at $\varepsilon$ (%) | | | | | |
|---|---|---|---|---|---|---|---|---|
| Diff. | Class. | ACR | 0.00 | 0.25 | 0.50 | 0.75 | 1.00 | 1.25 |
| ✗ | ✗ | 0.639 | 75.3 | 62.0 | 50.6 | 39.3 | 31.0 | 22.9 |
| ✓ | ✗ | 0.664 | 75.1 | 64.1 | 53.3 | 42.1 | **32.5** | 25.0 |
| ✗ | ✓ | 0.664 | **76.0** | **64.7** | 52.7 | 42.1 | 32.4 | 24.4 |
| ✓ | ✓ | **0.673** | 75.2 | **64.7** | **53.5** | **43.8** | **32.5** | 25.8 |

Table 5: Comparison of model error rates on CIFAR-10 decomposed into (a) over-smoothing ($p \leq p_0$) and (b) over-confidence ($p > p_0$), also with ACR.

| | Error rates (%, ↓) | | |
|---|---|---|---|
| Model ($\sigma = 1.0$) | $p \leq p_0$ | $p > p_0$ | ACR (↑) |
| Carlini et al. [11] | 43.8 | **7.6** | 0.498 |
| **Cascading** | 5.0 | 14.5 | 0.579 |
| **+ Anti-consist.** | 4.7 | 11.4 | 0.566 |
| **+ Brier loss** | **3.6** | 16.8 | **0.645** |

inputs. Specifically, in Table 1 and 2, we consider a fixed standard classifier of CLIP-finetuned ViT-B/16 on CIFAR-10 (or ImageNet), and compare the empirical accuracy of smoothed classifiers on CIFAR-10, -10-C [27] and -10.1 [54] (or ImageNet, -R [28], and -A [30]). Overall, the results show that our smoothed models could consistently outperform even in terms of empirical accuracy while maintaining high certified accuracy, *i.e.*, they abstain only when necessary. We additionally observe in Appendix C.2 that the empirical accuracy of our smoothed model can be further improved by considering an "ensemble" with the clean classifier: *e.g.*, the ensemble improves the accuracy of our cascaded classifier ($\sigma \in \{0.25, 0.5\}$) on CIFAR-10-C by $88.8\% \rightarrow 95.0\%$, even outperforming the accuracy of the standard classifier of $93.4\%$.

## 4.2 Ablation study

In Table 3, we compare our result with other training baselines as well as some ablations for a component-wise analysis, particularly focusing on their performance in cascaded smoothing across $\sigma \in \{0.25, 0.50, 1.00\}$ on CIFAR-10. Here, we highlight several remarks from the results, and provide the more detailed study, *e.g.*, the effect of $p_0$ (5), in Appendix C.3.

**Cascading from other training.** "Gaussian" and "Consistency" in Table 3 report the certified robustness of cascaded classifiers where each of single-scale models is individually trained by the method. Even while their (certified) clean accuracy of $\sigma = 0.25$ models are competitive with those of denoising-based models, their collective accuracy significantly degraded after cascading, and interestingly the drop is much more significant on "Consistency", although it did provide more robustness at larger $\varepsilon$. Essentially, for a high clean accuracy in cascaded smoothing the individual classifiers should make consistent predictions although their confidence may differ: the results imply that individual training of classifiers, without a shared denoiser, may break this consistency. This supports an architectural benefit of denoised smoothing for a use as cascaded smoothing.

**Cross-entropy *vs.* Brier.** "+ Cross-entropy" in Table 3 considers an ablation of the Brier loss (7) where the loss is replaced by the standard cross-entropy: although it indeed improves ACR compared to Carlini et al. [11] and achieves the similar clean accuracy with "+ Brier loss", its gain in overall certified robustness is significantly inferior to that of Brier loss as also reflected in the worse ACR. The superiority of the "squared" loss instead of the "log" loss suggests that it may not be necessary to optimize the confidence of individual denoised image toward a value strictly close to 1, which is reasonable considering the severity of noise usually considered for randomized smoothing.

**Anti-consistency loss.** The results marked as "+ Anti-consist." in Table 3, on the other hand, validates the effectiveness of the anti-consistency loss (8). Compared to "+ Brier loss", which is equivalent to the case when $\alpha = 0.0$ in (9), we observe that adding anti-consistency results in a slight decrease in ACR but with an increase in clean accuracy. The increased accuracy of cascaded smoothing indicates that the fine-tuning could successfully reduce over-confidence, and this tendency continues at larger $\alpha = 2.0$. Even with the decrease in ACR over the Brier loss, the overall ACRs attained are still superior to others, confirming the effectiveness of our proposed loss in total (9).

**Classifier fine-tuning.** In Table 4, we compare our proposed diffusion fine-tuning (Section 3.3) with another possible scheme of *classifier* fine-tuning, of which the effectiveness has been shown by Carlini et al. [11]. Overall, we observe that fine-tuning both classifiers and denoiser can bring their complementary effects to improve denoised smoothing in ACR, and the diffusion fine-tuning itself could obtain a comparable gain to the classifier fine-tuning. The (somewhat counter-intuitive) effectiveness of diffusion fine-tuning confirms that denoising process can be also biased as well as classifiers to make over-confident errors. We further remark two aspects where classifier fine-tuning can be less practical, especially when it comes with the cascaded smoothing pipeline we propose:

(a) To perform cascaded smoothing at multiple noise scales, classifier fine-tuning would require separate runs of training for optimal models per scale, also resulting in multiple different models to load - which can be less scalable in terms of memory efficiency.

(b) In a wider context of denoised smoothing, the classifier part is often assumed to be a model at scale, even covering cases when it is a "black-box" model, *e.g.*, public APIs. Classifier fine-tuning, in such cases, can become prohibitive or even impossible.

With respect to (a) and (b), denoiser fine-tuning we propose offers a more efficient inference architecture: it can handle multiple noise scales jointly with a single diffusion model, while also being applicable to the extreme scenario when the classifier model is black-box.

**Over-smoothing and over-confidence.** As proposed in Section 3.1, errors from smoothed classifiers can be decomposed into two: *i.e.*, $p \leq p_0$ for over-smoothing, and $p > p_0$ over-confidence, respectively. In Table 5, we further report a detailed breakdown of the model errors on CIFAR-10, assuming $\sigma = 1.0$. Overall, we observe that over-smoothing can be the major source of errors especially at such a high noise level, and our proposed cascading dramatically reduces the error. Although we find cascading could increase over-confidence from accumulating errors through multiple inferences, our diffusion fine-tuning could alleviate the errors jointly reducing the over-smoothing as well.

## 5 Conclusion and Discussion

Randomized smoothing has been traditionally viewed as a somewhat less-practical approach, perhaps due to its cost in inference time and impact on accuracy. In another perspective, to our knowledge, it is currently one of a few existing approaches that is prominent in pursuing *adversarial robustness at scale*, *e.g.*, in a paradigm where training cost scales faster than computing power. This work aims to make randomized smoothing more practical, particularly concerning on scalable scenarios of robustifying large pre-trained classifiers. We believe our proposals in this respect, *i.e.*, *cascaded smoothing* and *diffusion calibration*, can be a useful step towards building safer AI-based systems.

**Limitation.** A practical downside of randomized smoothing is in its increased inference cost, mainly from the majority voting procedure per inference. Our method also essentially possesses this practical bottleneck, and the proposed multi-scale smoothing scheme may further increase the cost from taking multiple smoothed inferences. Yet, we note that randomized smoothing itself is equipped with many practical axes to reduce its inference cost by compensating with abstention: for example, one can reduce the number of noise samples, *e.g.*, to $n = 100$. It would be an important future direction to explore practices for a better trade-off between the inference cost and robustness of smoothed classifiers, which could eventually open up a feasible way to obtain adversarial robustness at scale.

**Broader impact.** Deploying deep learning based systems into the real-world, especially when they are of security-concerned [12, 72], poses risks for both companies and customers, and we researchers are responsible to make this technology more reliable through research towards *AI safety* [2, 29]. *Adversarial robustness*, that we focus on in this work, is one of the central parts of this direction, but one should also recognize that adversarial robustness is still a bare minimum requirement for reliable deep learning. The future research should also explore more diverse notions of AI Safety to establish a realistic sense of security for practitioners, *e.g.*, monitoring and alignment research, to name a few.

## Acknowledgments and Disclosure of Funding

This work was partly supported by Center for Applied Research in Artificial Intelligence (CARAI) grant funded by Defense Acquisition Program Administration (DAPA) and Agency for Defense Development (ADD) (UD230017TD), and by Institute of Information & communications Technology Planning & Evaluation (IITP) grant funded by the Korea government (MSIT) (No.2019-0-00075, Artificial Intelligence Graduate School Program (KAIST)). We are grateful to the Center for AI Safety (CAIS) for generously providing compute resources that supported a significant portion of the experiments conducted in this work. We thank Kyuyoung Kim and Sihyun Yu for the proofreading of our manuscript, and Kyungmin Lee for the initial discussions. We also thank the anonymous reviewers for their valuable comments to improve our manuscript.

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

# A  Related Work

In the context of aggregating multiple smoothed classifiers, several works have previously explored on ensembling smoothed classifiers of the same noise scale [46, 32, 71], in attempt to boost their certified robustness. Our proposed multi-scale framework for randomized smoothing considers a different setup of aggregating smoothed classifiers of different noise scales, with a novel motivation of addressing the current accuracy-robustness trade-off in randomized smoothing. More in this respect, Mueller et al. [51] have considered a selective inference scheme between a "certification" network (for robustness) and a "core" network (for accuracy), and Horváth et al. [33] have adapted the framework into randomized smoothing. Our approach explores an orthogonal direction, and can be viewed as directly improving the "certification" network here: in this way, we could offer improvements not only in empirical but also in *certified* accuracy as a result.

Independently to our method, there have been approaches to further tighten the certified lower-bound that randomized smoothing guarantees [50, 70, 43, 45, 69]. Mohapatra et al. [50] have shown that higher-order information of smoothed classifiers, *e.g.*, its gradient on inputs, can tighten the certification of Gaussian-smoothed classifiers, beyond that only utilizes the zeroth-order information of smoothed confidence. In the context of denoised smoothing [58, 11], Xiao et al. [69] have recently proposed a multi-step, multi-round scheme for diffusion models to improve accuracy of denoising at randomized smoothing. Such approaches could be also incorporated to our method to further improve certification. Along with the developments of randomized smoothing, there have been also other attempts in certifying deep neural networks against adversarial examples [23, 68, 49, 25, 76, 16, 4, 77], where a more extensive survey on the field can be found in Li et al. [44].

# B  Experimental Details

## B.1  Datasets

**CIFAR-10** [39] consist of 60,000 images of size 32×32 pixels, 50,000 for training and 10,000 for testing. Each of the images is labeled to one of 10 classes, and the number of data per class is set evenly, *i.e.*, 6,000 images per each class. By default, we do not apply any data augmentation but the input normalization with mean $[0.5, 0.5, 0.5]$ and standard deviation $[0.5, 0.5, 0.5]$, following the standard training configurations of diffusion models. The full dataset can be downloaded at `https://www.cs.toronto.edu/~kriz/cifar.html`.

**CIFAR-10-C** [27] are collections of 75 replicas of the CIFAR-10 test datasets (of size 10,000), which consists of 15 different types of common corruptions each of which contains 5 levels of corruption severities. Specifically, the datasets includes the following corruption types: (a) *noise*: Gaussian, shot, and impulse noise; (b) *blur*: defocus, glass, motion, zoom; (c) *weather*: snow, frost, fog, bright; and (d) *digital*: contrast, elastic, pixel, JPEG compression. In our experiments, we evaluate test errors on CIFAR-10-C for models trained on the "clean" CIFAR-10 datasets, where the error values are averaged across different corruption types per severity level. The full datasets can be downloaded at `https://github.com/hendrycks/robustness`.

**CIFAR-10.1** [54] is a reproduction of the CIFAR-10 test set that are separately collected from Tiny Images dataset [62]. The dataset consists of 2,000 samples for testing, and designed to minimize distribution shift relative to the original CIFAR-10 dataset in their data creation pipelines. The full dataset can be downloaded at `https://github.com/modestyachts/CIFAR-10.1`, where we use the "v6" version for our experiments among those given in the repository.

**ImageNet** [56], also known as ILSVRC 2012 classification dataset, consists of 1.2 million high-resolution training images and 50,000 validation images, which are labeled with 1,000 classes. As a data pre-processing step, we apply a 256×256 center cropping for both training and testing images after re-scaling the images to have 256 in their shorter edges, making it compatible with the pre-processing of ADM [18], the backbone diffusion model. Similarly to CIFAR-10, all the images are normalized with mean $[0.5, 0.5, 0.5]$ and standard deviation $[0.5, 0.5, 0.5]$. A link for downloading the full dataset can be found in `http://image-net.org/download`.

**ImageNet-R** [28] consists of 30,000 images of various artistic renditions for 200 (out of 1,000) ImageNet classes: *e.g.*, art, cartoons, deviantart, graffiti, embroidery, graphics, origami, paintings, patterns, plastic objects, plush objects, sculptures, sketches, tattoos, toys, video game renditions,

and so on. To perform an evaluation of ImageNet classifiers on this dataset, we apply masking on classifier logits for the 800 classes those are not in ImageNet-R. The full dataset can be downloaded at `https://github.com/hendrycks/imagenet-r`.

**ImageNet-A** [30] consists of 7,500 images of 200 ImageNet classes those are "adversarially" filtered, by collecting natural images that causes wrong predictions from a pre-trained ResNet-50 model. To perform an evaluation of ImageNet classifiers on this dataset, we apply masking on classifier logits for the 800 classes those are not in ImageNet-A. The full dataset can be downloaded at `https://github.com/hendrycks/natural-adv-examples`.

## B.2 Training

**CLIP fine-tuning.** We follow the training configuration suggested by FT-CLIP [19] in fine-tuning the CLIP ViT-B/16@224 model, where the full script is specified at `https://github.com/LightDXY/FT-CLIP`. The same configuration is applied to all the datasets considered for fine-tuning, namely CIFAR-10, ImageNet, and Gaussian-corrupted versions of CIFAR-10 (for baselines such as "Gaussian" and "Consistency"), which can be done by resizing the datasets into $224 \times 224$ as a data pre-processing. We fine-tune all these tested models for 50 epochs on each of the considered datasets.

**Diffusion fine-tuning.** Similarly to the CLIP fine-tuning, we follow the training details given by Nichol and Dhariwal [52] in fine-tuning diffusion models, which are specified in `https://github.com/openai/improved-diffusion`. More concretely, here we fine-tune each pre-trained diffusion model by "resuming" the training following its configuration, but with an added regularization term we propose upon the original training loss. We use the 50M-parameter $32 \times 32$ diffusion model from Nichol and Dhariwal [52] for CIFAR-10,[6] and the 552M-parameter $256 \times 256$ unconditional model from Dhariwal and Nichol [18] for ImageNet[7] to initialize each fine-tuning run. We fine-tune each model for 50K training steps, using batch size 128 and 64 for CIFAR-10 and ImageNet, respectively.

## B.3 Hyperparameters

Unless otherwise noted, we use $p_0 = 0.5$ for *cascaded smoothing* throughout our experiments. We mainly consider two configurations of cascaded smoothing: (a) $\sigma \in \{0.25, 0.50, 1.00\}$, and (b) $\sigma \in \{0.25, 0.50\}$, each of which denoted as "(0.25-1.00)" and "(0.25-0.50)" in tables, respectively. For *diffusion calibration*, on the other hand, we use $\alpha = 1.0$ by default. We use $\lambda = 0.01$ on CIFAR-10 and $\lambda = 0.005$ on ImageNet, unless noted: we halve the regularization strength $\lambda$ on ImageNet considering its batch size used in the fine-tuning, *i.e.*, 128 (for CIFAR-10) *vs.* 64 (for ImageNet). In other words, we follow $\lambda = 0.01 \cdot (\texttt{batch\_size}/128)$. Among the baselines considered, "Consistency" [35] requires two hyperparameters: the coefficient for the consistency term $\eta$ and the entropy term $\gamma$. Following those considered by Jeong and Shin [35], we fix $\gamma = 0.5$ and $\eta = 5.0$ throughout our experiments.

## B.4 Computing infrastructure

Overall, we conduct our experiments with a cluster of 8 NVIDIA V100 32GB GPUs and 8 instances of a single NVIDIA A100 80GB GPU. All the CIFAR-10 experiments are run on a single NVIDIA A100 80GB GPU, including both the diffusion fine-tuning and the smoothed inference procedures. For the ImageNet experiments, we use 8 NVIDIA V100 32GB GPUs per run. In our computing environment, it takes *e.g.*, $\sim$10 seconds and $\sim$5 minutes per image (we use $n = 10,000$ for each inference) to perform a single pass of smoothed inference on CIFAR-10 and ImageNet, respectively. For a single run of fine-tuning diffusion models, we observe $\sim$1.5 days of training cost on CIFAR-10 with a single NVIDIA A100 80GB GPU, and that of $\sim$4 days on ImageNet using a cluster of 8 NVIDIA V100 32GB GPUs, to run 50K training steps.

At inference time, our proposed cascaded smoothing may introduce an extra computational overhead as the cost for the increased accuracy, *i.e.*, from taking multiple times of smoothed inferences. Nevertheless, remark that the actual overhead per input in practice can be different depending on the early-stopping result of cascaded smoothing. For example, in our experiments, we observe that

---

[6] `https://github.com/openai/improved-diffusion`
[7] `https://github.com/openai/guided-diffusion`

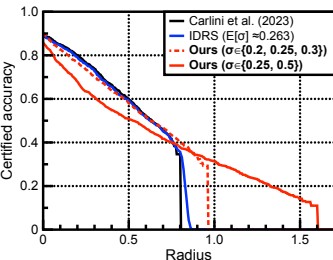
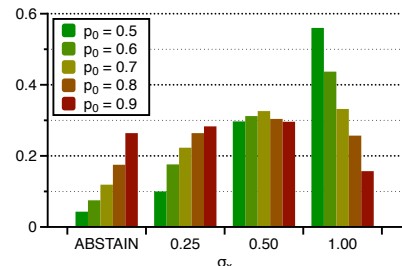

Figure 4: Comparison of certified test accuracy with (a) Carlini et al. [11] at $\sigma = 0.25$, (b) *Input-Dependent Randomized Smoothing* (IDRS) proposed by Súkeník et al. [60]. For (b), we adopt the per-sample $\sigma$ values of CIFAR-10 test set of Súkeník et al. [60].

Figure 5: Comparison of histograms across $p_0 \in [0.5, 1.0)$ on the distribution of $\sigma$'s where each of CIFAR-10 test samples halts from cascaded smoothing. Each of smoothed inferences is performed with $n = 10,000$ noise samples and $\alpha = 0.001$.

the average inference time to process the CIFAR-10 test set via cascaded smoothing is only $\sim 1.5\times$ compared to standard (single-scale) smoothed inferences, even considering $\sigma \in \{0.25, 0.5, 1.0\}$.

## C  Additional Results

### C.1  Comparison with input-dependent smoothing

In a sense that the proposed cascaded smoothing results in different smoothing factors $\sigma$ per-sample basis, our approach can be also viewed in the context of *input-dependent randomized smoothing* [1, 67, 13, 20], a line of attempts to improve certified robustness by applying different smoothing factor $\sigma$ conditioned on input. From being more flexible in assigning $\sigma$, however, these approaches commonly require more assumptions for their certification, making them hard to compare directly upon the standard evaluation protocol, *e.g.*, that we considered in our experiments. For example, Súkeník et al. [60] have proved that such a scheme of freely assigning $\sigma$ does not provide valid robustness certificates in the standard protocol, and [1, 20] in response have suggested to further assume that all the previous test samples during evaluation can be memorized to predict future samples, which can affect the practical relevance of the overall evaluation protocol.

To address the issue, Súkeník et al. [60] have also proposed a fix of [1] to discard its test-time dependency, by assuming a more strict restriction on the variation of $\sigma$. Nevertheless, they also report that the benefit of input-dependent $\sigma$ then becomes quite limited, supporting the challenging nature of input-dependent smoothing. As shown in Figure 4 that additionally compares our method with [60], we indeed confirm that cascaded smoothing offers significantly wider robustness certificates with a better accuracy trade-off. This is unprecedented to the best of our knowledge, which could be achieved from our completely new approach of data-dependent smoothing: *i.e.*, by combining multiple smoothed classifiers through abstention, moving away from the previous attempts that are focused on obtaining a single-step smoothing.

### C.2  Clean ensemble for empirical accuracy

In this appendix, we demonstrate that predictions from smoothed classifiers can not only useful to obtain adversarial robustness, but also to improve out-of-distribution robustness of *clean* classifiers, which suggests a new way to utilize randomized smoothing. Specifically, consider a smoothed classifier $\hat{f}$, and its smoothed confidences at $\mathbf{x}$, say $p_{\hat{f}}(y|\mathbf{x})$ for $y \in \mathcal{Y}$. Next, we consider prediction confidences of a "clean" prediction, namely $p_f(y|\mathbf{x})$. Remark that $p_f(y|\mathbf{x})$ can be interpreted in terms of *logits*, namely $\log p_f(\mathbf{x}|y)$, of a neural network based model $f$ as follows:

$$p_f(y|\mathbf{x}) = \frac{p_f(\mathbf{x}, y)}{p_f(\mathbf{x})} = \frac{p(y) \cdot p_f(\mathbf{x}|y)}{\sum_{y \in \mathcal{Y}} p_f(\mathbf{x}, y)}. \tag{10}$$

Note that the prior $p(y)$ is usually assumed to be the uniform distribution $\mathcal{U}(\mathcal{Y})$. Here, we consider a variant of this prediction, by replacing the prior $p(y)$ by a *mixture* of $p(y)$ and a *smoothed prediction*

Table 6: Comparison of per-corruption empirical accuracy of clean ("FT-CLIP") and smoothed classifiers (Others) on CIFAR-10-C. For each corruption type, we report accuracy averaged over 5 levels of severity, and for each level we use 1,000 uniform subsamples to compute the accuracy.

| Method | Clean | Gaussian | Shot | Impulse | Defocus | Glass | Motion | Zoom | Snow | Frost | Fog | Brightness | Contrast | Elastic | Pixelate | JPEG | AVG |
|---|---|---|---|---|---|---|---|---|---|---|---|---|---|---|---|---|---|
| FT-CLIP [19] | **98.1** | 83.7 | 88.7 | **96.4** | **96.8** | 85.7 | 95.0 | 96.1 | **96.1** | 96.0 | **97.0** | **97.8** | **97.5** | 94.7 | 91.0 | 88.5 | 93.4 |
| Carlini et al. [11] | 88.8 | 85.8 | 86.9 | 88.7 | 87.3 | 83.6 | 86.8 | 86.0 | 85.2 | 83.6 | 78.9 | 87.8 | 75.7 | 85.4 | 86.4 | 85.8 | 84.9 |
| + Casc. + Calib. | 92.5 | 89.0 | 90.1 | 91.5 | 90.7 | 87.3 | 89.6 | 90.7 | 89.9 | 89.4 | 86.0 | 91.1 | 81.2 | 89.7 | 90.4 | 89.3 | 89.1 |
| + Clean prior | 98.0 | **90.9** | **93.0** | 96.5 | **96.8** | **89.9** | **95.2** | **96.3** | **96.2** | **96.6** | 96.8 | 97.7 | 97.4 | **95.3** | **94.2** | **91.6** | **95.0** |

Table 7: Comparison of empirical accuracy and certified accuracy of cascaded smooth classifiers on CIFAR-10 across ablations on $\lambda$. We use $\sigma \in \{0.25, 0.5, 1.0\}$ for cascaded smoothing.

| CIFAR-10 | | Empirical | Certified accuracy at $\varepsilon$ (%) | | | | | | | |
|---|---|---|---|---|---|---|---|---|---|---|
| Setups | | Clean | 0.00 | 0.25 | 0.50 | 0.75 | 1.00 | 1.25 | 1.50 | 1.75 |
| $\lambda = 0.001$ | $\alpha = 1.0$ | 84.1 | 79.9 | 54.4 | 39.6 | 30.9 | 24.2 | 18.9 | 13.9 | 9.3 |
| | $\alpha = 2.0$ | 84.3 | 80.7 | 53.9 | 38.9 | 29.7 | 23.9 | 18.7 | 13.0 | 9.6 |
| | $\alpha = 4.0$ | 85.1 | 80.9 | 52.8 | 39.1 | 29.5 | 23.2 | 17.3 | 12.4 | 9.4 |
| | $\alpha = 8.0$ | 86.1 | 81.5 | 53.2 | 38.5 | 29.0 | 22.0 | 17.1 | 13.0 | 8.8 |
| $\lambda = 0.01$ | $\alpha = 1.0$ | 82.9 | 79.6 | 53.5 | 40.5 | 31.2 | 25.0 | 20.1 | 16.2 | 11.4 |
| | $\alpha = 2.0$ | 84.8 | 80.1 | 54.6 | 39.2 | 30.6 | 24.0 | 18.5 | 14.5 | 9.9 |
| | $\alpha = 4.0$ | 85.1 | 80.4 | 53.6 | 39.1 | 29.5 | 22.8 | 17.8 | 13.1 | 9.5 |
| | $\alpha = 8.0$ | 86.1 | 81.5 | 54.5 | 39.0 | 27.7 | 21.4 | 17.3 | 12.9 | 8.8 |
| $\lambda = 0.1$ | $\alpha = 1.0$ | 80.8 | 77.2 | 53.9 | 39.9 | 31.4 | 24.8 | 20.1 | 15.5 | 11.6 |
| | $\alpha = 2.0$ | 82.7 | 78.0 | 53.9 | 39.9 | 30.7 | 24.1 | 19.8 | 15.1 | 11.1 |
| | $\alpha = 4.0$ | 82.1 | 78.3 | 52.6 | 39.1 | 30.1 | 22.6 | 17.4 | 13.7 | 10.0 |
| | $\alpha = 8.0$ | 83.7 | 78.2 | 51.1 | 36.9 | 28.0 | 22.3 | 17.9 | 13.1 | 9.1 |

Table 8: Comparison of empirical accuracy and certified accuracy of cascaded smooth classifiers on CIFAR-10 across ablations on $p_0$. We use $\sigma \in \{0.25, 0.5, 1.0\}$ for cascaded smoothing.

| CIFAR-10 | Empirical | Certified accuracy at $\varepsilon$ (%) | | | | | | | |
|---|---|---|---|---|---|---|---|---|---|
| Setups | Clean | 0.00 | 0.25 | 0.50 | 0.75 | 1.00 | 1.25 | 1.50 | 1.75 |
| $p_0 = 0.50$ | 84.8 | 81.1 | 52.8 | 37.2 | 27.9 | 21.7 | 16.8 | 11.8 | 8.5 |
| $p_0 = 0.55$ | 87.9 | 82.6 | 53.5 | 35.6 | 26.9 | 19.8 | 14.0 | 10.1 | 7.1 |
| $p_0 = 0.60$ | 90.7 | 83.9 | 52.7 | 34.5 | 23.9 | 16.9 | 11.9 | 8.3 | 6.4 |
| $p_0 = 0.65$ | 92.5 | 83.4 | 52.1 | 34.0 | 22.6 | 14.4 | 9.8 | 7.0 | 5.5 |
| $p_0 = 0.70$ | 94.1 | 83.0 | 52.6 | 31.6 | 19.2 | 12.2 | 8.2 | 6.3 | 4.7 |
| $p_0 = 0.75$ | 95.2 | 81.9 | 52.2 | 29.1 | 16.5 | 9.5 | 6.7 | 5.2 | 3.7 |
| $p_0 = 0.80$ | 96.0 | 79.5 | 47.6 | 25.0 | 14.0 | 8.0 | 5.9 | 4.6 | 2.4 |
| $p_0 = 0.85$ | 96.6 | 76.3 | 45.0 | 22.8 | 12.5 | 7.1 | 4.7 | 2.8 | 2.2 |
| $p_0 = 0.90$ | 97.2 | 72.1 | 42.5 | 20.0 | 11.7 | 4.6 | 2.8 | 1.9 | 1.0 |
| $p_0 = 0.95$ | 97.8 | 65.5 | 38.6 | 16.7 | 7.2 | 2.2 | 1.7 | 0.7 | 0.0 |

$p(\hat{y}) := p_{\hat{f}}(y|\mathbf{x})$: in this way, one could use $\hat{f}(\mathbf{x})$ as an additional prior to refine the clean confidence in cases when the two sources of confidence do not match. More specifically, we consider:

$$\log p_f(\hat{y}|\mathbf{x}) := \log p_{\hat{f}}^{\beta}(y|\mathbf{x}) + \log p_f(\mathbf{x}|y) + C, \quad \text{where} \tag{11}$$

$$\log p_{\hat{f}}^{\beta}(y|\mathbf{x}) := (1-\beta) \cdot \log p_{\hat{f}}(y|\mathbf{x}) + \beta \cdot p(y). \tag{12}$$

where $C$ denotes the normalizing constant, and $\beta > 0$ is a hyperparameter which we fix as $\beta = 0.1$ in our experiment. In Table 6, we compare the accuracy of the proposed inference with the accuracy of standard classifiers derived from FT-CLIP [19] fine-tuned on CIFAR-10: overall, we observe that this "ensemble" of clean prediction with cascaded smoothed predictions could improve the corruption accuracy of FT-CLIP on CIFAR-10-C, namely by $93.4\% \rightarrow 95.0\%$, while maintaining its clean accuracy. The gain can be considered significant in a sense that the baseline accuracy of $93.4\%$ is already at the state-of-the-art level, *e.g.*, according to RobustBench [17].

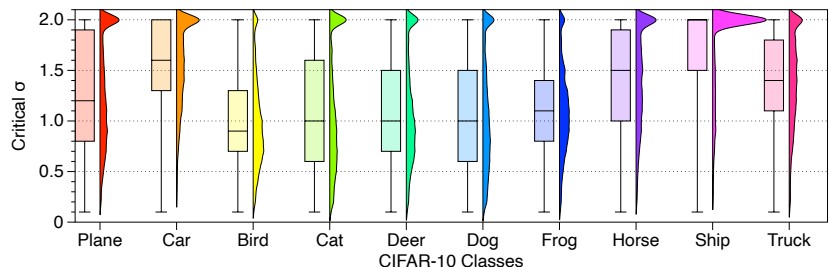

Figure 6: Comparison of per-class histograms of *critical* $\sigma$: the minimum value of $\sigma$ for each input where the smoothed confidence at true class falls below $0.5$. We use the whole CIFAR-10 training samples for the analysis. Each image is smoothed with $n = 1,000$ samples of Gaussian noise, while iterating $\sigma$ within the range $\sigma \in [0.0, 2.0]$. Images with critical $\sigma$ larger than $2.0$ are truncated.

## C.3  Ablation study

**Effect of $\lambda$.**  In Table 7, we jointly ablate the effect of tuning $\lambda \in \{0.001, 0.01, 0.1\}$ upon the choices of $\alpha \in \{1.0, 2.0, 4.0, 8.0\}$, by comparing its certified accuracy of cascaded smoothing as well as their empirical accuracy on CIFAR-10 test set. Overall, we observe that increasing $\lambda$ has an effect of improving robustness at large radii, with a slight decrease in the clean accuracy as well as the empirical accuracy, where using higher $\alpha$ could compensate this at some extent. Although we generally observe effectiveness of the proposed regularization for a wide range of $\lambda$, using higher values of $\lambda$, *e.g.*, $\lambda = 0.1$, often affects the base diffusion model and results in a degradation in accuracy without a significant gain in its certified robustness.

**Effect of $p_0$.**  Table 8, on the other hand, reports the effect of $p_0$ we introduced for cascaded smoothing. Recall that we use $p_0 = 0.5$ throughout our experiments by default. We observe that using a bit of higher values of $p_0$, *e.g.*, $p_0 = 0.6$ could be beneficial to improve the clean accuracy of cascaded smoothing, at the cost of robustness at larger radii: for example, it improves the certified accuracy at $r = 0.0$ by $81.1 \rightarrow 83.9$. The overall certified robustness starts to decrease as $p_0$ further increases, due to its effect of increasing abstention rate. One could still observe the consistent improvement in the empirical accuracy as $p_0$ increases, *i.e.*, by also considering abstain as a valid prediction, *i.e.*, using high values of $p_0$ could be useful in certain scenarios. We think a more detailed choice of $p_0$ per $\sigma$ can further boost the performance of cascaded smoothing, which is potentially an important practical consideration.

**Critical $\sigma$.**  Although it is not required for cascaded smoothing to guarantee the stability of smoothed confidences across $\sigma$ (*e.g.*, outside of $\{0.25, 0.5, 1.0\}$ in our experiments), we observe that smoothed confidences from denoised smoothing usually interpolate smoothly between different $\sigma$'s, at least for the diffusion denoised smoothing pipeline. In this respect, we consider a new concept of *critical $\sigma$* for each sample, by measuring the threshold of $\sigma$ where its confidence goes below $p_0 = 0.5$. Figure 6 examines this measure on CIFAR-10 training samples and plots its distribution for each class as histograms. Interestingly, we observe that the distributions are often significantly biased, *e.g.*, the "Ship" class among CIFAR-10 classes relatively obtains much higher critical $\sigma$ - possibly due to its peculiar background of, *e.g.*, ocean.

**Per-$\sigma$ distribution from cascading.**  To further verify that per-sample $\sigma$'s from cascaded smoothing are diverse enough to achieve better robustness-accuracy trade-off, we examine in Figure 5 the actual histograms of the resulting $\sigma \in \{\texttt{ABSTAIN}, 0.25, 0.50, 1.00\}$ on CIFAR-10, also varying $p_0 \in \{0.5, 0.6, 0.7, 0.8, 0.9\}$. Overall, we observe that (a) the distributions of $\sigma_x$ indeed widely cover the total range of $\sigma$ for every $p_0$'s considered, and (b) using higher $p_0$ can further improve diversity of the distribution while its chance to be abstained can be also increased.

## D  Technical Details

### D.1  Proof of Theorem 3.1

Our proof is based on the fact that any smoothed classifier $\hat{f}$ composed with $\Phi^{-1}(\cdot)$ is always Lipschitz-continuous, which is shown by Salman et al. [57]:

**Lemma D.1** (followed by Salman et al. [57]). *Let $\Phi(a) := \frac{1}{\sqrt{2\pi}} \int_{-\infty}^{a} \exp\left(-\frac{1}{2}s^2\right) \mathrm{d}s$ be the cumulative distribution function of standard Gaussian. For any function $f : \mathbb{R}^d \to [0,1]$, the map $\mathbf{x} \mapsto \sigma \cdot \Phi^{-1}(p_{\hat{f}_\sigma}(\mathbf{x}))$ is 1-Lipschitz for any $\sigma > 0$.*

Given Lemma D.1, the robustness certificate by Cohen et al. [15] in (2) (of the main text) can be viewed as computing the region of $p_{\hat{f}_\sigma}(\mathbf{x} + \boldsymbol{\delta}; y) > 0.5$, *i.e.*, where it is guaranteed to return $y$ over other (binary) smoothed classifiers $p_{\hat{f}_\sigma}(\mathbf{x} + \boldsymbol{\delta}; y')$ of $y \neq y'$.

In general, as in our proposed *cascaded smoothing*, one can consider a more "strict" region defined by $p_{\hat{f}_\sigma}(\mathbf{x} + \boldsymbol{\delta}; y) > p_0$ for some $p_0 \geq 0.5$. Here, the region corresponds to a policy that $\hat{f}_\sigma$ abstains whenever $p_{\hat{f}_\sigma}(\mathbf{x}) \leq p_0$. Accordingly, the certified radius one can obtain from this policy becomes:

$$R_{p_0}(\hat{f}_\sigma; \mathbf{x}, y) := \sigma \cdot \left( \Phi^{-1}(p_{\hat{f}_\sigma}(\mathbf{x}, y)) - \Phi^{-1}(p_0) \right). \tag{13}$$

With this notation, we restate Theorem 3.1 into a tighter form to show, which do not consider neither $\underline{p}$ nor $\overline{p}$ upon the knowledge of $p_{\hat{f}_{\sigma_k}}$ and implies Theorem 3.1 as a consequence:

**Theorem 3.1 (restated).** *Let $\hat{f}_{\sigma_1}, \cdots, \hat{f}_{\sigma_K} : \mathcal{X} \to \mathcal{Y}$ be smoothed classifiers with $0 < \sigma_1 < \cdots < \sigma_K$. Suppose $\mathtt{casc}(\mathbf{x}; \{\hat{f}_{\sigma_i}\}_{i=1}^K) =: \hat{y} \in \mathcal{Y}$ halts at $\hat{f}_{\sigma_k}$ for some $k$. Then, it holds that $\mathtt{casc}(\mathbf{x} + \boldsymbol{\delta}; \{\hat{f}_{\sigma_i}\}_{i=1}^K) = \hat{y}$ for $\|\boldsymbol{\delta}\| < R$, where:*

$$R := \min\left\{ R_{p_0}(\hat{f}_{\sigma_k}; \mathbf{x}, \hat{y}), R^- \right\}, \text{ and } R^- := \min_{\substack{y \neq \hat{y} \\ k' > k}} \left\{ \min\left\{ 0, -R_{p_0}(\hat{f}_{\sigma_{k'}}; \mathbf{x}, y) \right\} \right\}. \tag{14}$$

*Proof.* By the definition of $\mathtt{casc}(\mathbf{x})$, one can ensure $\mathtt{casc}(\mathbf{x} + \boldsymbol{\delta}) = \hat{y}$ if the following holds:

(a) $\hat{f}_{\sigma_k}(\mathbf{x} + \boldsymbol{\delta}) = \hat{y}$ with $p_{\hat{f}_{\sigma_k}}(\mathbf{x} + \boldsymbol{\delta}, \hat{y}) > p_0$, and

(b) $p_{\hat{f}_{\sigma_{k'}}}(\mathbf{x} + \boldsymbol{\delta}) < p_0$ or $\hat{f}_{\sigma_{k'}}(\mathbf{x} + \boldsymbol{\delta}) = \hat{y}$, for every $k' = k + 1, \cdots, K$.

Remark that the condition (a) is satisfied by $\boldsymbol{\delta}$ within $\|\boldsymbol{\delta}\| < R_{p_0}(\hat{f}_{\sigma_k}; \mathbf{x}, \hat{y})$. Specifically, one can consider the "certifiable" region of $\boldsymbol{\delta}$ at $\hat{f}_{\sigma_k}$ as follows:

$$\mathcal{R}_k = \{ \boldsymbol{\delta} \in \mathcal{X} : \|\boldsymbol{\delta}\| < R_{p_0}(\hat{f}_{\sigma_k}; \mathbf{x}, \hat{y}) \}. \tag{15}$$

For the condition (b), on the other hand, note that the condition is equivalent to that:

(b′) $p_{\hat{f}_{\sigma_{k'}}}(\mathbf{x} + \boldsymbol{\delta}; y) < p_0$ for $y \neq \hat{y}$, and for every $k' = k + 1, \cdots, K$.

Fix $k' \in \{k + 1, \cdots, K\}$. From the Lipschitzedness of $\mathbf{x} \mapsto \sigma \cdot \Phi^{-1}(p_{\hat{f}_\sigma}(\mathbf{x}; y))$ by Lemma D.1, it is clear to show that $\boldsymbol{\delta}$'s inside the following region $\mathcal{R}_{k'}$ satisfies (b′):

$$\mathcal{R}_{k'} = \bigcap_{y \neq \hat{y}} \left\{ \boldsymbol{\delta} \in \mathcal{X} : \|\boldsymbol{\delta}\| < \min\left\{ 0, -R_{p_0}(\hat{f}_{\sigma_{k'}}; \mathbf{x}, y) \right\} \right\} \tag{16}$$

$$= \left\{ \boldsymbol{\delta} \in \mathcal{X} : \|\boldsymbol{\delta}\| < \min_{y \neq \hat{y}} \left\{ \min\left\{ 0, -R_{p_0}(\hat{f}_{\sigma_{k'}}; \mathbf{x}, y) \right\} \right\} \right\}. \tag{17}$$

Lastly, the result is followed by considering the intersection of $\mathcal{R}_k$'s which represent a certifiable region where $\mathtt{casc}(\mathbf{x} + \boldsymbol{\delta}; \{\hat{f}_{\sigma_i}\}_{i=1}^K) = \hat{y}$:

$$\mathcal{R} := \mathcal{R}_k \cap (\cap_{k' \geq k+1} \mathcal{R}_{k'}) = \{ \boldsymbol{\delta} \in \mathcal{X} : \|\boldsymbol{\delta}\| < R \}. \tag{18}$$

$\square$

Once we have the restated version of Theorem 3.1 above, the original statement follows from an observation that the new radius by introducing either $\underline{p}$ or $\overline{p}$, say $\underline{R}$, always lower-bounds $R$ of the above: *i.e.*, $\underline{R} \leq R$ for any choices of $\underline{p}$ and $\overline{p}$, thus certifiable for the region as well.

## D.2 Certification and prediction

For practical uses of cascaded smoothing, where the exact values of $p_{\hat{f}_{\sigma_k}}(\mathbf{x}, y)$ are not available, one has to estimate the smoothed confidence values as per Theorem 3.1, *i.e.*, by properly lower-bound (or upper-bound) the confidence values. More concretely, one is required to obtain (a) $\underline{p}_{\hat{f}_{\sigma_k}}(\mathbf{x})$ at the halting stage $k$, and (b) $\overline{p}_{\hat{f}_{\sigma_{k'}}}(\mathbf{x}, y)$ for $k' = k + 1, \cdots, K$ and $y \in \mathcal{Y}$.

For both estimations, we use Monte Carlo algorithm with $n$ *i.i.d.* Gaussian samples, say $\{\boldsymbol{\delta}_i\}$, and perform a majority voting across $\{f_\sigma(\mathbf{x} + \boldsymbol{\delta}_i)\}$ to obtain a histogram on $\mathcal{Y}$ of $n$ trials:

(a) To estimate a lower-bound $\underline{p}_{\hat{f}_{\sigma_k}}(\mathbf{x})$, we follow the statistical procedure by Cohen et al. [15], namely CERTIFY$(n, n_0, \alpha)$, which additionally takes two hyper-parameters $n_0$ and $\alpha$: here, $n_0$ denotes the number of samples to initially make a guess to estimate a smoothed prediction, and $\alpha$ is the significance level of the Clopper-Pearson confidence interval [14]. Concretely, it sets $\underline{p}_{\hat{f}_{\sigma_k}}$ as the lower confidence level of $p_{\hat{f}_{\sigma_k}}$ with coverage $1 - \alpha$.

(b) For $\overline{p}_{\hat{f}_{\sigma_{k'}}}(\mathbf{x}, y)$, on the other hand, we first obtain voting counts from $n$ Gaussian trials, say $\mathbf{c}_\mathbf{x}^{(\sigma_{k'})}$, and adopt Goodman confidence interval to bound multinomial proportions [24] with the significance level $\alpha$. Specifically, one can set $\overline{p}_{\hat{f}_{\sigma_{k'}}}(\mathbf{x}, y)$ for $y \in \mathcal{Y}$ by the $(1 - \alpha)$-upper level of the (multinomial) confidence interval. This can be implemented by using, *e.g.*, `multinomial_proportions_confint` of the `statsmodels` library.

We make an additional treatment on the significance level $\alpha$: unless the pipeline halts at the first stage $K$, cascaded smoothing makes a $(K - k + 1)$-consecutive testing of confidence intervals to make a prediction. In attempt to maintain the overall significance level of this prediction as $\alpha$, one can apply the Bonferroni correction [5] on the significance level of individual testing, specifically by considering $\frac{\alpha}{(K-k+1)}$ as the adjusted significance level (instead of using $\alpha$) at the stage $k$.

# E  Other Possible Designs

In this appendix, we present two alternative designs of multi-scale randomized smoothing we have also considered other than *cascaded smoothing* proposed in the main text. Specifically, here we additionally propose two different policies of aggregating multiple smoothed classifiers, each of which dubbed as the (a) *max-radius policy*, and (b) *focal smoothing*, respectively. Overall, both design do provide valid certified guarantees combining the accuracy-robustness trade-offs across multiple smoothing factors, often attaining a better robustness even compared to cascaded smoothing when $n$, the number of noise samples, is large enough.

The common drawback in the both approaches here, however, is the necessity to access every smoothed confidence, which can be crucial in practice in both terms of efficiency and accuracy. Specifically, for given $K$ smoothed classifiers, this implies the exact $K$-times increase in the inference cost, which makes it much costly on average compared to cascaded smoothing. The need to access and compare with $K$ smoothed confidence in practice also demands more accurate estimations of the confidence values to avoid abstaining. For example, these approaches can significantly degrade the certified accuracy in practice where $n$ cannot be large unlike the certification phase: which again makes our cascaded smoothing more favorable over the approaches presented here.

## E.1  Max-radius policy

Consider $K$ smoothed classifiers, say $\hat{f}_{\sigma_1}, \cdots, \hat{f}_{\sigma_K} : \mathcal{X} \to \mathcal{Y}$, where $0 < \sigma_1 < \cdots < \sigma_K$. The *max-radius* policy, say $\mathtt{maxR}(\cdot)$, can be another simple policy one can consider for given $\{\hat{f}_{\sigma_i}\}$:

$$\mathtt{maxR}(\mathbf{x}; \{\hat{f}_{\sigma_i}\}_{i=1}^K) := \hat{f}_{\sigma^*}(\mathbf{x}), \text{ where } \sigma^* := \underset{\sigma_i}{\arg\max} \; \sigma_i \cdot \Phi^{-1}\left(p_{\hat{f}_{\sigma_i}}(\mathbf{x})\right). \qquad (19)$$

Again, given the Lipschitzedness associated with the smoothed confidences $p_{\hat{f}_\sigma}(\mathbf{x}, y)$, it is easy to check the following robustness certificate for $\mathtt{maxR}(\mathbf{x}; \{\hat{f}_{\sigma_i}\}_{i=1}^K)$:

**Theorem E.1.** *Let $\hat{f}_{\sigma_1}, \cdots, \hat{f}_{\sigma_K} : \mathcal{X} \to \mathcal{Y}$ be smoothed classifiers with $0 < \sigma_1 < \cdots < \sigma_K$. If $\mathtt{maxR}(\mathbf{x}; \{\hat{f}_{\sigma_i}\}_{i=1}^K) =: \hat{y} \in \mathcal{Y}$, it holds that $\mathtt{maxR}(\mathbf{x} + \boldsymbol{\delta}; \{\hat{f}_{\sigma_i}\}_{i=1}^K) = \hat{y}$ for $\|\boldsymbol{\delta}\| < R$, where:*

$$R := \frac{1}{2} \cdot \left( \sigma^* \cdot \Phi^{-1}\left(p_{\hat{f}_{\sigma^*}}(\mathbf{x})\right) - \max_{\substack{y \neq \hat{y} \\ i=1,\cdots,K}} \sigma_i \cdot \Phi^{-1}\left(p_{\hat{f}_{\sigma_i}}(\mathbf{x};y)\right) \right). \tag{20}$$

Remark that the certified radius given by (20) recovers the guarantee of Cohen et al. [15] if $K = 1$.

## E.2 Focal smoothing

In this method, coined *focal smoothing*, we take a somewhat different approach to achieve multi-scale smoothing. Here, now we assume two different levels of smoothing factor, say $\sigma_0 > \sigma_1 > 0$, and consider inputs that contains noise with *spatially different* magnitudes. Specifically, we consider a binary mask $M \in \{0, 1\}^d$ and an input corrupted by the following:

$$\mathtt{focal}(\mathbf{x}, M; \sigma_0, \sigma_1) := \mathbf{x} + (\sigma_0 \cdot (1 - M) + \sigma_1 \cdot M) \odot \boldsymbol{\epsilon}, \tag{21}$$

where $\odot$ denotes the element-wise product and $\boldsymbol{\epsilon} \sim \mathcal{N}(0, \mathbf{I})$. In this way, each of corrupted input $\hat{\mathbf{x}}$ gets a "hint" to recover the original content of $\mathbf{x}$ by observing a less-corrupted region marked by $M$.

Randomized smoothing over such *anisotropic* noise has been recently studied [22, 20]: in a nutshell, smoothing over noises specified by (21) could produce an *ellipsoid-shaped* certified region with two axes of length $\sigma_0 \Phi^{-1}(p)$ and $\sigma_1 \Phi^{-1}(p)$ for its smoothed confidence $p$, respectively.

In terms of certified radius, however, this implies that one can still certify only the region limited by the shorter axes (of length $\sigma_1 \Phi^{-1}(p)$). It turns out that such a limitation can be avoided by considering an "ensemble" of $K$ multiple orthogonal masks, say $M_1, \cdots, M_K$, so that the shorter axes of ellipsoids from the masks do not overlap each other. Specifically, we observe the following:

**Theorem E.2.** *Let $M_1, \cdots, M_K \in \{0, 1\}^d$ be $K$ orthogonal masks and $\sigma_0 > \sigma_1 > 0$ be smoothing factors, and $p_k(\mathbf{x})$ be the smoothed confidence from inputs corrupted by $\mathtt{focal}(\mathbf{x}, M_k; \sigma_0, \sigma_1)$, for a given binary classifier $f_k : \mathbb{R}^d \to [0, 1]$. Define $\hat{p} := \frac{1}{K} \sum_k p_k$, and suppose $\hat{p}(\mathbf{x}) > 0.5$. Then, it is certified that $\hat{p}(\mathbf{x} + \boldsymbol{\delta}) > 0.5$ for $\|\boldsymbol{\delta}\| < R_{\mathbf{b}^*}$, where $\mathbf{b}^*$ is the solution of the following optimization:*

$$\begin{aligned} \underset{\mathbf{b}}{\text{minimize}} \quad & R_{\mathbf{b}} := \frac{1}{K} \sum_k \frac{a_k}{\sqrt{t^2 b_k + 1}} \\ \text{subject to} \quad & \sum_k b_k = 1, \\ & b_k \geq 0, \ k = 1, 2, \cdots, K, \end{aligned} \tag{22}$$

*where $a_k := \sigma_0 \Phi^{-1}(p_k)$ for $k = 1, \cdots, K$, and $t^2 := \left(\frac{\sigma_0^2}{\sigma_1^2} - 1\right)$.*

In Algorithm 1, we propose an efficient 1-dimensional grid-search based optimization to solve (22), based on observations given as Lemma E.3 and E.4: each of which identifies the condition of $b_k$ when it attains either the maximum value or zero, respectively. By solving the optimization, one could obtain a certified radius $R$ that the aggregated smoothing can guarantee consistent prediction.

**Lemma E.3.** *Consider the constrained optimization (22) with respect to $\mathbf{b} = (b_1, \cdots, b_K)$. Let $M_{\mathbf{a}} := \{m \in [K] : a_m = \max_k a_k\}$ and $\mathbf{b}^*$ be an optimal solution of (22). Then, it holds that $b_m^* = b_{m'}^* > 0$ for any $m, m' \in M_{\mathbf{a}}$.*

*Proof.* First, we show that there exists at least one $m \in M_{\mathbf{a}}$ where $b_m^*$ is nonzero, *i.e.*, $b_m^* > 0$. Suppose the contrary, *i.e.*, $b_k^* = 0$ for all $k \in M_{\mathbf{a}}$. Denote the corresponding objective value as $R^*$. Given the condition $\sum_k b_k = 1$, there always exists $i \notin M$ where $b_i > 0$. Now, consider swapping the allocations of $b_m$ and $b_i$ for some $m \in M_{\mathbf{a}}$ and denote the corresponding objective value $R'$.

**Algorithm 1** Focal smoothing: A grid-search based optimization of (22)

---
**Require:** $a_k > 0$, $k = 1, 2, \ldots, K$, tolerance $\varepsilon > 0$
**Ensure:** optimal budget assignment $\mathbf{b}^*$, certified radius $R^*$
  1: Find the maximum value $a_{\max} = \max_k a_k$
  2: Identify the index set $M$ such that $a_k = a_{\max}$ for $k \in M$
  3: Set a grid of values for $b_m$ in the interval $(0, 1]$.
  4: **for** each grid value of $b_m$ **do**
  5:   **for** $k = 1, 2, \ldots, K$ **do**
  6:     **if** $k \notin M$ **then**
  7:       $b_k \leftarrow 0$ **if** $a_k \leq \frac{a_m}{(t^2 b_m + 1)^{3/2}}$ **else** $\frac{1}{t^2}\left(\left(\frac{a_k}{a_m}\right)^{2/3}(t^2 b_m + 1) - 1\right)$
  8:     **end if**
  9:   **end for**
 10:   **if** $|\sum_{k=1}^K b_k - 1| \leq \varepsilon$ **then**
 11:     $R \leftarrow \frac{1}{K}\sum_{k=1}^K \frac{a_k}{\sqrt{t^2 b_k + 1}}$
 12:     Update $\mathbf{b}_k^*, R^* \leftarrow b_k, R$ if $R < R^*$
 13:   **end if**
 14: **end for**

---

Then, we have:

$$
\begin{aligned}
R' - R^* &= \frac{1}{K}\left(\frac{a_k}{\sqrt{t^2 b_j + 1}} + \frac{a_j}{\sqrt{t^2 b_k + 1}} - \frac{a_k}{\sqrt{t^2 b_k + 1}} - \frac{a_j}{\sqrt{t^2 b_j + 1}}\right) \\
&= \frac{1}{K}\left(a_k\left(\frac{1}{\sqrt{t^2 b_j + 1}} - \frac{1}{\sqrt{t^2 b_k + 1}}\right) + a_j\left(\frac{1}{\sqrt{t^2 b_k + 1}} - \frac{1}{\sqrt{t^2 b_j + 1}}\right)\right) \\
&< \frac{1}{K}\left(a_k\left(\frac{1}{\sqrt{t^2 b_j + 1}} - \frac{1}{\sqrt{t^2 b_k + 1}}\right) + a_k\left(\frac{1}{\sqrt{t^2 b_k + 1}} - \frac{1}{\sqrt{t^2 b_j + 1}}\right)\right) = 0,
\end{aligned}
\tag{23}
$$

given that $a_k > a_j$ and $\frac{1}{\sqrt{t^2 b_j + 1}} < \frac{1}{\sqrt{t^2 b_k + 1}} = 1$. This means $R' < R^*$, contradicting the optimality of the initial solution $\mathbf{b}^*$ with $b_k = 0$ for all $k \in M_{\mathbf{a}}$. Thus, there must exist at least one $b_m^* > 0$ for some $m \in M_{\mathbf{a}}$.

Next, we show $b_m^* = b_{m'}^*$ for any $m, m' \in M_{\mathbf{a}}$. Suppose there exist such $m, m' \in M_{\mathbf{a}}$ where $b_m^* \neq b_{m'}^*$. Due to the symmetry, one can construct another optimal allocation $\mathbf{b}'$ by swapping the two allocations $b_m^*$ and $b_{m'}^*$, as it does not affect the optimality. Now consider another allocation $\bar{\mathbf{b}}$ by averaging the two allocations:

$$
\bar{b}_m = \bar{b}_{m'} = \frac{b_m^* + b_{m'}^*}{2}.
\tag{24}
$$

Let the corresponding objective value $\bar{R}$. Note that $R$ is strictly convex, which it implies that $\bar{R} < R^*$, contradicting the optimality of the initial solution $\mathbf{b}^*$. Thus, it holds $b_m^* = b_{m'}^*$ for any $m, m' \in M_{\mathbf{a}}$. $\qquad\square$

**Lemma E.4.** *Consider the constrained optimization* (22) *with* $\mathbf{b}^*$ *as an optimal solution. Denote* $B := b_m^* = b_{m'}^* > 0$ *for all* $m, m' \in M_{\mathbf{a}}$*. Then,* $b_k^* = 0$ *if and only if* $a_k \leq \frac{A}{(t^2 B + 1)^{3/2}}$*.*

*Proof.* Suppose $b_k^* = 0$. From the Karush–Kuhn–Tucker (KKT) stationarity condition, we have $\frac{-a_k t^2}{K} + \lambda - \mu_k = 0$. Since $\lambda, \mu_k \geq 0$, we obtain $\lambda \geq \frac{a_k t^2}{K}$. Now, consider the stationarity condition for $m \in M$: $-\frac{At^2}{K(\sqrt{t^2 B + 1})^3} + \lambda = 0$. Substituting the lower bound for $\lambda$, we have $a_k \leq \frac{A}{(t^2 B + 1)^{3/2}}$. Conversely, suppose $a_k \leq \frac{A}{(t^2 B + 1)^{3/2}}$. Then, $b_k^* > 0$ contradict to that $b_k^* = \frac{1}{t^2}\left(\left(\frac{a_k}{A}\right)^{2/3}(t^2 B + 1) - 1\right) \leq 0$, hence $b_k^* = 0$. $\qquad\square$

