# OpenReview forum: "Multi-scale Diffusion Denoised Smoothing"
_NeurIPS.cc/2023/Conference — NeurIPS 2023 poster_

### Official Review · Reviewer_rNFT · 2023-06-25

**Soundness:** 3 good
**Presentation:** 3 good
**Contribution:** 2 fair
**Rating:** 5
**Confidence:** 5

**Summary:**

This paper proposed a new technique for certified adversarial robustness based on randomized smoothing. Two certification schemes are proposed in the paper: first is cascaded randomized smoothing where multiple smoothed classifiers of different smoothing factors are aggregated together in an efficient way; the second part is a new fine-tuning method called diffusion calibration to address the overconfidence issue in the original randomized smoothing approach. Experimental results show that the proposed method improved upon the previous baseline approach of Diffusion Denoised Smoothing.

**Strengths:**

1. Clarity: This paper is well-written. The method is clearly described and introduced. The results are presented in a concise and clear fashion.
2. Originality: The method is novel in two aspects. First, the proposed scheme of cascading smoothed classifiers of multiple smoothing factors is novel and promising as an approach for combining multiple smoothed classifiers. Second, two losses are proposed for calibrating the diffusion model in Diffusion Denoised Smoothing.
3. Significance: Randomized smoothing and its variant, for example, diffusion denoised smoothing are known to have practical limitations as they require huge certification samples to compute the certification bound. The proposed method for more efficient certification could pave the way for practical applications of randomized smoothing and diffusion denoised smoothing.

**Weaknesses:**

1. While the proposed method delivers improvement of certification accuracies upon the standard diffusion denoised smoothing, the paper does not discuss the computational requirements. It would be good to discuss the computational overhead from running a cascading smoothed classifier and fine-tuning the diffusion denoiser, in the main paper.
2. It seems from the description of cascaded randomized smoothing in section 3.2, the prediction of the cascaded smoothed classifier is defined, different from what the original randomized smoothing proposed. Then i think there should be a subsection or paragraph defining the certification bounds from the cascaded smoothed classifier. Theorem 1 does not seem to point to which one is the new certification bound. I would suggest the authors to add a pseudocode in the main text or appendix describing how the prediction and certification is done for the proposed method, in a similar way to what the original randomized smoothing paper [1] did.
3. Section 3.1 discussed some of the limitations of randomized smoothing. It is not clear whether this is a contribution of this work or has been observed before. From the way it is described, these limitations seems like well-recognized in the research community. If that’s the case, it would be better to put this part in the background section to avoid confusion about the novelty.
4. Regarding the specific limitations of randomized smoothing discussed in Section 3.1, is there evidence that supports these arguments? It would be good to show empirical evidence of overconfidence or point to a previous work that has looked into this matter.
5. The Brier loss and Anti-consistency loss are proposed to address the overconfidence issue in randomized smoothing. The experiments could benefit from adding an evaluation confirming that the overconfidence issue is tackled via the proposed fine-tuning process, thus further supporting the validity and effectiveness of the approach.
6. The main contributions of the work are two parts: cascaded randomized smoothing and diffusion calibration. However, the title “multi-scale diffusion denoised smoothing” seems to only focus on the first part of the contribution. I would recommend a more proper title to reveal the true contributions of this work, which are not limited to cascaded randomized smoothing.

[1] Certified adversarial robustness via randomized smoothing. Cohen et al, 2019.

==================================================================
I have read the rebuttal and i will remain my score.

**Questions:**

See Weaknesses.

**Limitations:**

Yes

---

> ### Author Rebuttal · Authors · 2023-08-09
>
> We sincerely appreciate your thoughtful comments, efforts, and time. We respond to each of your questions one-by-one in what follows. Please let us know if you have any comments/concerns that we have not addressed up to your satisfaction.
>
> ---
>
> **Q1. Computational overhead**
>
> Thank you for the suggestion, and we will report the respective discussion in the final draft. As you mentioned, our proposed cascaded smoothing may introduce an inference overhead as the cost for an increased accuracy, i.e., from taking multiple times of smoothed inferences. Nevertheless, we remark that the actual overhead per input depends on the early-stopping in cascaded smoothing in practice: for example, in our experiments, we observe that the average inference time to process the CIFAR-10 test set via cascaded smoothing ($\sigma\in \\{0.25, 0.5, 1.0\\}$) is only ~1.5x compared to the standard one. We will incorporate this and more results in the final draft.
>
> ---
>
> **Q2. Detailed certification procedure**
>
> Indeed, our implementation of cascaded smoothing has a treatment to make it statistically correct across _i.i.d._ noise samples, considering a certain _significance level_ $\alpha$ (e.g., $\alpha = 0.001$ in our experiments) in a similar fashion as Cohen et al. [1]. This certification procedure is described in Appendix C.2 for your information, and we will further provide more details in the final draft, e.g., by adding a pseudocode of the procedure as per your suggestion.
>
> ---
>
> **Q3. Section 3.1: Is this a prior observation, or a contribution?**
>
> We believe that (a) our re-interpretation of smoothed confidence, especially in the context of diffusion denoised smoothing, and (b) the corresponding decomposition of model errors based on it can be novel contributions to the community. While smoothed confidence has been used to abstain inputs in randomized smoothing, it has primarily been seen as a trick to ensure statistical guarantee of certified radius. It is from our observations to develop methods to address model errors separately based on types, i.e., through cascaded smoothing and diffusion fine-tuning. We will make this point clearer in the final draft.
>
> ---
>
> **Q4. Empirical support of over-confidence and over-smoothing**
>
> For a given smoothed classifier, one can indeed measure the two types of model errors following their definitions we consider in this work, i.e., $p \le p_0$ for over-smoothing and $p > p_0$ over-confidence, respectively. Following your suggestion, we will further report and compare this detailed breakdown of model error, e.g., as provided in Table S2 of the rebuttal PDF: overall, we observe that over-smoothing can be the major source of errors especially at high noise level, e.g., $\sigma=1.0$, and our proposed cascading dramatically reduces the error. Although we find cascading could increase over-confidence from accumulating errors through multiple inferences, our diffusion fine-tuning could alleviate the errors jointly reducing the over-smoothing as well. We will include these and more results in the final draft, as well as the respective discussions.
>
> ---
>
> **Q5. Title**
>
> Thank you for the suggestion. We agree that there can be a better title which can further emphasize our key contributions: cascaded smoothing and diffusion calibration. We have chosen the current title to succinctly reflect our key aspect, but we will definitely consider your suggestion and revisit our title, while ensuring it continues to be concise.
>
> ---
>
> **References**
>
> [1] Cohen et al., Certified adversarial robustness via randomized smoothing, ICML 2019.

---

### Official Review · Reviewer_K2e2 · 2023-07-01

**Soundness:** 3 good
**Presentation:** 3 good
**Contribution:** 3 good
**Rating:** 6
**Confidence:** 5

**Summary:**

This paper presents a practical method to address the issues of over-smoothing and over-confidence in randomized smoothing, thereby enhancing our understanding of the trade-off between accuracy and robustness. For over-smoothing, the author introduces a multi-stage approach for computing collective certified robustness without the need to specify noise scales. This novel process provides a more precise evaluation of the certified radius. Additionally, to address over-confidence, this study incorporates two proposed loss functions, namely Brier loss and Anti-consistency loss, to fine-tune the denoiser model. By integrating these two methods into Diffusion Denoised Smoothing, significant improvements in both clean and certified accuracy can be achieved.

**Strengths:**

1. Cascaded randomized smoothing is a specifically designed method for more accurate certified radius evaluation for Diffusion Denoised Smoothing because with diffusion models, we do not need to train smoothed classifiers for each noise scale.

2. Evaluate both empirical accuracy and certified accuracy for the robustness of the proposed method.

**Weaknesses:**

1. I don't think the cascaded randomized smoothing can overcome the trade-off between accuracy and certified robustness. Cascaded randomized smoothing is only a tricky strategy to make a better computation of the certified radius. It cannot improve certified robustness intrinsically.

2. For calibrating diffusion models, I don't see an advantage compared with classifier fine-tuning in Carlini et al. And Table 2 in Carlini et al. shows a better performance than yours with classifier finetuning. Besides, normally speaking, fine-tuning classifiers is much simpler than fine-tuning diffusion models.

**Questions:**

1. Results shown in your table are different from those shown in the original paper of Carlini et al. For example, in Table 1, the highest clean acc is 78.8 while the original paper shows 88.1. Is that because you use a different classifier? If yes, why you do not choose to use the classifier used in Carlini et al.? They can even reach the better results.

2. In Table 3, if the clean acc of Carlini et al. is higher than your results with Anti-consist, how can you demonstrate that your proposed loss can improve clean accuracy? I think it can only improve the overall clean accuracy with your cascaded strategy.

**Limitations:**

No limitations about potential negative societal impact.

---

> ### Author Rebuttal · Authors · 2023-08-09
>
> We sincerely appreciate your thoughtful comments, efforts, and time. We respond to each of your questions one-by-one in what follows. Please let us know if you have any comments/concerns that we have not addressed up to your satisfaction.
>
> ---
>
> **Q1. “Cascaded smoothing cannot improve certified robustness intrinsically.”**
>
> We agree that the “intrinsic” approach you mentioned to enhance adversarial robustness, e.g., presumably by training a more robust model representation, should be an important path forward - indeed, this is why we have also explored diffusion fine-tuning as well as cascaded smoothing in our work.
>
> Nevertheless, we politely disagree that such an intrinsic approach is the only way to improve the robustness-accuracy trade-off, especially in the context of randomized smoothing where the trade-off is often “unavoidable”. For example, Table 1 of our paper reports that even the current state-of-the-art, e.g., Carlini et al. [1], could only achieve the clean accuracy $<50\\%$ on CIFAR-10 to obtain robustness at $\varepsilon=2.0$: in order to obtain a non-zero robustness at $\varepsilon = 2.0$ with randomized smoothing, one has to perturb a given input with severe Gaussian noise of $\sigma = 1.0$ and it often makes some inputs inevitably non-classifiable.
>
> In this respect, we view our proposal of cascaded smoothing as not only a simple strategy to optimize certified radius, but also a novel attempt to overcome the trade-off fundamentally induced by the data itself. In the aforementioned example, one can now get a chance to reduce the noise level, e.g., from $\sigma = 1.0$ to $\sigma = 0.5$, to turn a “non-classifiable” input “classifiable”, while maintaining the robustness of other samples. Knowing the right amounts of per-input adversarial budget $\varepsilon$ for better generalization has been one of open questions not only in the context of randomized smoothing but also in adversarial machine learning at large, e.g., [2], and we believe our approach could be a useful step to answer the question.
>
> ---
>
> **Q2. Fine-tuning denoiser vs. classifier**
>
> Thank you for your comment. We have addressed this specific comment in the common response, so please refer to the section for a detailed answer.
>
> ---
>
> **Q3. Table 1: Lower baseline results compared to Carlini et al. [1]?**
>
> We clarify that there might be some confusion regarding the results presented: the clean accuracy rate you mentioned, 78.8%, is actually from our results in Table 2, which is based on ImageNet. The corresponding result for Table 1, i.e., on CIFAR-10, is 89.5%, which is actually better than the 88.1% from the original paper by Carlini et al. [1]. In general, the clean accuracy of denoised smoothing tends to follow that of the backbone classifier, and this could explain why the highest accuracy in Table 1 is better than those reported in Carlini et al.: specifically, our choice of classifier performs better (98.1% on CIFAR-10) compared to theirs (97.9%).
>
> ---
>
> **Q4. Table 3: Clean accuracy is degraded?**
>
> We remark that a comparison of smoothed classifiers is not solely about clean accuracy: it is important to consider the trade-off between certified robustness and accuracy. This is a reason why we have adopted average certified radius (ACR) as a key metric in our comparisons. The Brier loss we propose is primarily to enhance certified robustness, which might occasionally come at the cost of slightly reduced accuracy. Even with the inclusion of anti-consistency, there could be a minor decline in accuracy. Yet, given the substantial improvements in ACR shown in both ablations, we believe any decrease in accuracy can be easily compensated, e.g., by reducing the relative strength of the Brier loss.
>
> As another note, we also highlight that the decrease in accuracy displayed in Table 3 is not a constant across different datasets: e.g., for the ImageNet results presented in Table 2, we observe that our proposed fine-tuning not only improved overall certified robustness but also clean accuracy.
>
> ---
>
> **References**
>
> [1] Carlini et al., (Certified!!) Adversarial Robustness for Free!, ICLR 2023.
>
> [2] Tramer et al., Fundamental Tradeoffs between Invariance and Sensitivity to Adversarial Perturbations, ICML 2020.

---

> > ### Comment · Reviewer_K2e2 · 2023-08-16
> >
> > The authors have solved most of my concerns, so I increased the score to 6.

---

> > > ### Author Response · Authors · 2023-08-17
> > > **Thank you for the feedback**
> > >
> > > Dear Reviewer K2e2,
> > >
> > > Thank you for your response! We are delighted to hear that our response letter could address your concerns. Your insightful comments will be integrated to make our manuscript even stronger.
> > >
> > > Best regards,
> > >
> > > Authors

---

### Official Review · Reviewer_b42a · 2023-07-02

**Soundness:** 3 good
**Presentation:** 3 good
**Contribution:** 3 good
**Rating:** 6
**Confidence:** 1

**Summary:**

This paper identifies the accuracy-robustness trade-off in randomized smoothing and the over-smoothing and over-confidence issues in diffusion denoised smoothing. It then proposes a to finetune diffusion models to mitigate these issues by first passing a datapoint through a cascaded smoothing pipeline to obtain class label and then minimizing Brier loss and Anti-consistency losses. This allows the framework to achieve higher empirical accuracy and robustness.

**Strengths:**

- The paper is clearly written and well motivated.
- The cascaded pipeline is novel and the explanations of the problems at hand are clear.
- Empirical evidence for improvement is strong.
- Ablation study and analysis is performed to investigate the effect of each proposed loss.

**Weaknesses:**

There are several places in the exposition that may require more clarification.

- Why do we not update classifier? Line 218 - 220 state that finetinung classifier is more computationally intensive than finetuning denoiser. Isn't backpropogating through an entire trajectory of denoising process more prohibitively expensive than finetuning classifier, especially when the starting noise-level is high?

- Can the classifier itself cause any of the two over-smoothing or over-confidence issues? In this case, do we wan to finetune classifiers? In so, how do we decide which model to finetune, or both?

- There lack some empirical investigation on the runtime issue with denoising, as is more apparent for such a long sequence of denoising iterations. The cascaded smoothing starts at a high noise-level, so denoising should take longer time. What's the trade-off between the accuracy/robustness increase with runtime?

**Questions:**

My concerns and questions are written above.

**Limitations:**

No limitations of the model are discussed. Inference time increase can be one to further discuss and analyze. Broader societal impacts are adequately addressed as improving adversarial robustness leads to safer AI systems.

---

> ### Author Rebuttal · Authors · 2023-08-09
>
> We sincerely appreciate your thoughtful comments, efforts, and time. We respond to each of your questions one-by-one in what follows. Please let us know if you have any comments/concerns that we have not addressed up to your satisfaction.
>
> ---
>
> **Q1. The higher $\sigma$, the longer runtime?**
>
> Thank you for pointing out an important detail we overlooked to specify. Here, we clarify that the runtime of the diffusion denoised smoothing pipeline [1] (that we are based on) does NOT depend on the choice of $\sigma$. This is because the pipeline is originally proposed to use “single-step” denoising (which is usually available from standard diffusion models) for its denoiser operation at any $\sigma$, based on the observation that it is beneficial both for efficiency as well as accuracy [1]. In our experiments, we have also followed this design choice. We will clarify this point in the final draft.
>
> ---
>
> **Q2. Fine-tuning denoiser vs. classifier**
>
> As also discussed in Q1, our proposed diffusion fine-tuning scheme does not require backpropagation through multiple denoising iterations, given that the denoiser operation in each smoothing is restricted to single-step. We have also addressed this specific question in the common response, so please refer to the section for a further discussion.
>
> ---
>
> **Q3. Limitations are not discussed, e.g., inference time?**
>
> For your information, we note that Appendix A does contain a discussion about limitations, indeed covering the increased inference time of our method. In the final draft, we will make this clearer in the main text as well.
>
> ---
>
> **References**
>
> [1] Carlini et al., (Certified!!) Adversarial Robustness for Free!, ICLR 2023.

---

### Official Review · Reviewer_QpH2 · 2023-07-06

**Soundness:** 3 good
**Presentation:** 3 good
**Contribution:** 3 good
**Rating:** 5
**Confidence:** 4

**Summary:**

This paper builds on diffusion based denoised randomized smoothing. It analyzes two scenarios of model errors which are over-smoothing and over-confidence. To alleviate these issues, the authors proposed deploying cascaded randomized smoothing where samples are first smoothed with large variance that gradually decreases if the model abstains from prediction. Further, the diffusion model (denoiser) is calibrated to overcome the over-confidence problem.

**Strengths:**

This work has the following strengths:

1- The problem this work is tackling is important. Building reliable and robust models are necessary for deploying such models.

2- The paper reads smoothly in most parts. Section 3.1 introduces two problems in randomized smoothing and Sections 3.2 and 3.3 propose the two components of the proposed approach.

3- The experiments conducted in this work covered the standard datasets and 3 related baselines.


**Weaknesses:**

Despite the strengths of this work, there are few weaknesses that I would like to be resolved before getting this paper accepted:

1- The experimental results are missing important strong baselines such as [1, 36] (from the main paper) and [12] from the appendix. Since the method of this work results in a different smoothing parameters for (potentially) different inputs, a comparison with data-dependent smoothing approaches is necessary.

2- The proposed approach seems to provide very limited performance gains in the small certified accuracy regimes. For instance, comparing the certified accuracy in Table 1 at $\sigma \in [0.25, 1.0]$ and $\epsilon =0.5$: The proposed method attains 40.5% while [8] attains a 59.8 using $\sigma = 0.25$. Shouldn't a fair comparison be comparing the proposed method to the envelop curve of each baseline across the three $\sigma$ values?

3- One ablation experiment that can potentially showcases the diversity of the obtained $\sigma$ via cascaded smoothing is to plot a histogram of $\sigma_x$ for when $\sigma \in [0.25, 1.0]$.

4- Another experimental analysis is to show the number of samples that were suffering from over-smoothing or over-confidence before employing the cascaded approach and after.

5- In terms of writing, I have the following minor comments/suggestions:
- The preliminary part about diffusion models in lines 116-128 is not used afterwards, especially the notation. Consider moving it to the appendix.
- $n$ was defined twice in the "Evaluation metrics" paragraph (line 285 and line 290) with two different values. Please consider modifying the second one.
- It is confusing to have more than one bold number per column in each table. Consider highlighting only one (perhaps the combined cascading + calibration).


**Questions:**

Please refer to the weaknesses part.

**Limitations:**

The authors have addressed the limitations in Appendix A.

---

> ### Author Rebuttal · Authors · 2023-08-09
>
> We sincerely appreciate your thoughtful comments, efforts, and time. We respond to each of your questions one-by-one in what follows. Please let us know if you have any comments/concerns that we have not addressed up to your satisfaction.
>
> ---
>
> **Q1. Comparison with previous data-dependent approaches?**
>
> We first clarify that most of previous data-dependent smoothing approaches, including [1-4], are not directly comparable upon the standard evaluation protocol we considered, due to their additional assumptions to ensure correctness given the theoretical challenge of data-dependent smoothing. For example, [1] and [4] require that all the previous test samples during evaluation can be memorized to predict future samples, which can affect the practical relevance of the overall evaluation protocol.
>
> To address the issue, Sukenik et al. [5] have recently proposed a fix of [1] to discard its test-time dependency: nevertheless, they also report that the benefit of data-dependent $\sigma$ becomes quite limited with the fix, supporting its challenging nature. Upon your suggestion, we have further performed an additional comparison of our method with [5], e.g., as in Figure S2 of the attached rebuttal PDF, confirming that our proposed cascaded smoothing offers significantly wider robustness certificates with a better accuracy trade-off - which is unprecedented to the best of our knowledge. This is achieved with our completely new approach of data-dependent smoothing, i.e., by combining multiple smoothed classifiers through abstention, moving away from previous attempts that are focused on obtaining a single-step smoothing. We will incorporate this and further results into the final draft, as well as the respective discussions.
>
> ---
>
> **Q2. Limited gains at small $\varepsilon$’s, e.g., compared to the $\sigma$-envelope?**
>
> The focus of our evaluations, e.g., in Table 1 and 2, is to compare smoothed classifiers taking into account their trade-off between certified robustness and accuracy. For example, although the certified accuracy of a smoothed classifier with $\sigma=0.25$ may perform better at small $\varepsilon$ compared to cascaded smoothing with $\sigma=[0.25, 1.0]$, the classifier cannot provide robustness certification at $\varepsilon \ge 1.0$. This is why we chose average certified radius (ACR) as a major performance metric in this paper, which is in compliance with the current literature: specifically, ACR assigns 0 for incorrect samples while giving more weights for samples with better robustness.
>
> Although the envelope curve across multiple $\sigma$’s can be a succinct proxy to evaluate different methods, it can be somewhat misleading and unfair to compare the curve directly with individual smoothed classifiers. This is because the curve does not really construct a concrete classifier on its own: specifically, it additionally assumes that each test sample has prior knowledge on the value of $\sigma \in  \\{0.25, 0.5, 1.0\\}$ to apply, which is itself challenging to infer - this is indeed what our proposal of cascaded smoothing addresses. We will clarify this point in the final draft.
>
> ---
>
> **Q3. Suggestions for ablation study**
>
> Thank you for your suggestions to improve our ablation study, viz., (a) the distribution of $\sigma_x$, and (b) the effect of cascading on over-smoothing, which would strengthen our manuscript. For (a), we have additionally examined the histogram of $\sigma \in \\{\mathtt{ABSTAIN}, 0.25, 0.50, 1.00\\}$ from cascaded smoothing on CIFAR-10 at varying $p_0$, e.g., as plotted in Figure S3 of the rebuttal PDF: overall, we observe that distributions of $\sigma_x$ widely cover across the range for all the $p_0$’s tested; For (b), on the other hand, we will further report and compare a more detailed breakdown of model errors with respect to the two categories, i.e., over-smoothing ($p \le p_0$) and over-confidence ($p > p_0$), e.g., as reported in Table S2 of the rebuttal PDF: here, e.g., we observe that over-smoothing can be the major source of errors especially at high noise level, e.g., $\sigma=1.0$, and our proposed cascading dramatically reduces the error. We will incorporate these results into the final draft, as well as more results, e.g., a qualitative inspection of samples with different $\sigma_x$’s.
>
> ---
>
> **Q4. Editorial comments**
>
> Many thanks for the careful reading to improve our manuscript. We will incorporate all of your editorial comments and suggestions into the final draft.
>
> ---
>
> **References**
>
> [1] Alfarra et al., Data dependent randomized smoothing, UAI 2022.
>
> [2] Wang et al., Pretrain-to-finetune adversarial training via sample-wise randomized smoothing, 2021.
>
> [3] Chen et al., Insta-RS: Instance-wise randomized smoothing for improved robustness and accuracy, 2021.
>
> [4] Eiras et al., ANCER: Anisotropic certification via sample-wise volume maximization, TMLR 2022.
>
> [5] Sukenik et al., Intriguing properties of input-dependent randomized smoothing, ICML 2022.

---

> > ### Comment · Reviewer_QpH2 · 2023-08-21
> >
> > Dear Authors
> >
> > I would like to thank you for the efforts put into the rebuttal. My remaining unresolved concern is the following.
> > While indeed both [1] and [4] (following the references in the rebuttal) employ a memory-based algorithm to ensure the soundness of the certificate, I still think it is very relevant to include a comparison between the proposed method and the ones in [1, 4] since both approaches yield different smoothing parameters for different inputs.
> >
> > That being said, I would like to keep my original (borderline accept) score.

---

> > > ### Author Response · Authors · 2023-08-21
> > > **Thank you for the feedback**
> > >
> > > Dear Reviewer QpH2,
> > >
> > > We appreciate your response, and the clarification on your remaining concern. Although the memory-based inference of [1] and [4] should require a further assumption to the current evaluation, i.e., test-time dependency as mentioned, we agree with you that having a more detailed comparison with [1, 4] can be still a good addition to improve the thoroughness of our manuscript. Following your suggestion, we will include additional comparisons of our method with [1, 4] in the final draft, as well as the respective discussions (we hope for your understanding that we could not update the results within the timeframe of this discussion phase).
> > >
> > > As another note, we would like to highlight again that the additional baseline [5] we compared in our earlier response can be already viewed as an instance of [1] - but with a more restriction on $\sigma$ to bypass the need of the memory-based inference. Given our significant improvements upon [5] as shown in Figure S2, we believe that our novel approach can be an important step towards developing data-dependent smoothing with a more practical assumption.
> > >
> > > Thank you again for your suggestion to improve our manuscript,
> > >
> > > Authors

---

> > > > ### Comment · Reviewer_QpH2 · 2023-08-22
> > > >
> > > > Dear Authors
> > > >
> > > > Thank you very much for the fast response.
> > > >
> > > > I understand that time restriction for providing such experimental comparison in the limited window of the rebuttal period. I am also aware that [5] employs a more conservative instance of [1] where the proposed method outperforms [5]. I want to note that although both [1, 4] depend on the order in which test data is presented, their experimental results showed small variance in the performance under different orders given the high dimensionality of the image datasets such as ImageNet. I do believe that such a comparison with strong baselines will enhance the experimental section in this work.
> > > >
> > > > Best,

---

> > > > > ### Author Response · Authors · 2023-08-22
> > > > > **Thank you for your response**
> > > > >
> > > > > Dear Reviewer QpH2,
> > > > >
> > > > > Thank you for your additional insight. Indeed, we see that the small variance of [1, 4] across different test sample orders can be an evidence to support the practical relevance of the methods despite their test-time dependency: we will also mention this point in the final draft. We also appreciate your understanding about the time constraints of this rebuttal period. As mentioned, we will work on conducting the additional comparisons and add them to further strengthen our manuscript.
> > > > >
> > > > > Best regards,
> > > > >
> > > > > Authors

---

### Official Review · Reviewer_Vnny · 2023-07-06

**Soundness:** 2 fair
**Presentation:** 1 poor
**Contribution:** 2 fair
**Rating:** 5
**Confidence:** 4

**Summary:**

This work aims to improve robustness-accuracy tradeoff in image classifiers, using the recent smoothing ideas. The authors propose two techniques for improving adversarial robustness without loss of accuracy. One is to fine-tune the prepended denoiser with a regularized loss, to reduce miss-classification of smoothed images. Second is to use a cascade of denoisers trained on different levels of noise to pick the highest level robustness radius. They test their method on CIFAR10 and ImageNet datasets.



**Strengths:**

1) Focusing on important topic of robustness of classifiers.
2) Interesting method in cascading models from high noise to low noise.
3) Good empirical results compared to recent state of the art models

**Weaknesses:**

The writing can be improved a lot both in terms of coherence and clarity. I found the paper lacking in a good flow and hard to read, because of redundant repetition, lack of a coarse to fine description of the goal and the result, redundant undefined notation. The authors The same amount of content can be written such that the reader can take the message with much less effort.

Whenever a new notation is introduced, please describe what it is defining. The reader should not have to infer what symbol is refering to what. They should be mentioned explicitly in the text. Examples: $\mathbb{P}, \epsilon, x', \underline{R}, \underline{p}
,\overline{p}, R^-$ ... This makes the paper more time consuming to read which is very unnecessary. As an example Theorem 3.1 is  unreadable due to undefined notation.

I find the use of "diffusion" in the title somewhat misleading. I suggest diffusion be replaced with denoising since that is what has been used in the work as opposed to a diffusion process in the context of generative models.

In line 109 it is mentioned that the ideal denoiser should return the original image for any noisy image. This statement is incorrect and hold only for very small levels of noise. The error in denoising is inherent in ideal denoisers as well and hence the iteration.

In order to reduce misclassification, the authors suggest fine-tuning the denoiser instead of the classifier to make the training less cumbersome. Although this resulted in better accuracy, it is not clear if it makes sense in theory. The cause of unrobustness is the random or incorrect boundaries set by the classifier. The denoiser is optimal with respect to denoising loss, whereas the classifier is not optimal with respect to classifying smoothed images (it has never been trained to do so). So in theory, the network that should be fine-tuned is the classifier not the denoiser.

**Questions:**

1) The idea of cascading denoise and classify models from large to small noise levels is a clever way to find the largest radius of robustness. The final sigma at which the model does not abstain and returns a classification output is taken as the certifies radius. This value however depends on two factors: one is the threshold $p_0$: for higher threshold the radius will be smaller and for lower threshold the radius will be larger. It is not clear in the text how this dependency is addressed. The second factor is the image content and the image class: for example in CIFAR10 for a class with generally distinctive background and content the certified radius will be larger (more smoothed images will be classified with high confidence). On the other hand, if the average background is similar between a few classes then the radius will be smaller. How do you address this variability?

2) For a fixed $p_0$ and a fixed image, do different noise samples (different runs of the method) result in different certified radius and or different predictions? If that is the case, how is this variability addressed?

3) Have you experimented with stability of prediction after the optimal radius is achieved? In other words, one way to quantify the error (or incorrect over confidence in the cascade) is to keep running the cascade for all the smaller noise levels and measure how volatile the prediction is. Assuming that for $\sigma=0$ the predication is correct (consistent with your oracle assumption on $f_{std}$), you can measure the $\sigma$ level for which the incorrect prediction flips to the correct one. Will you see a significant difference for different classes for this critical $sigma$?


**Limitations:**

I found the paper lacking solid theoretical ground. The methods, although clever, are ad hoc and do not address the robustness against adversarial attacks fundamentally. Regardless, due to impressive empirical results my score is borderline accept. I will increase the score if the authors address the comments and questions.

---

> ### Author Rebuttal · Authors · 2023-08-09
>
> We sincerely appreciate your thoughtful comments, efforts, and time. We respond to each of your questions one-by-one in what follows. Please let us know if you have any comments/concerns that we have not addressed up to your satisfaction.
>
> ---
>
> **Q1. Suggestions for better clarity - e.g., notation, title, etc.**
>
> Many thanks for your suggestions to improve the clarity of our manuscript. We will carefully revise our manuscript to reflect your comments, e.g., by adding more detailed descriptions of the mathematical notations.
>
> ---
>
> **Q2. Line 109: “ideal denoiser”?**
>
> We appreciate your detailed feedback on the term “ideal denoiser” we mentioned for an intuition of denoised smoothing. We agree that the notation $\texttt{denoise}(\mathbf{x} + \boldsymbol{\delta}) \approx \mathbf{x}$ could be more precise if it was further noted as “semantic” equivalence with respect to $f$, rather than the exact match. We will clarify this point in the final draft.
>
> ---
>
> **Q3. Fine-tuning denoiser vs. classifier**
>
> Thank you for your comment. We have addressed this specific comment in the common response, so please refer to the section for a detailed answer.
>
> ---
>
> **Q4. How is the effect of $p_0$ addressed?**
>
> We simply use a fixed value of $p_0 = 0.5$ in our experiments (as specified in Appendix B.3), but we note here for your information that we did perform an ablation study of $p_0$ in Appendix E.2. Indeed, the choice of threshold $p_0$ affects the certified radius of cascading, as well as its accuracy: higher $p_0$ tends to decrease the overall certified radii, but it may increase the accuracy due to the increased chance to reach lower $\sigma$. We think the detailed choice of $p_0$ per $\sigma$ can further boost the results, which would be an interesting practical consideration. We will further clarify this point in the final draft.
>
> ---
>
> **Q5. Variability of $\sigma$ over classes**
>
> As you mentioned, the distributions of output $\sigma$ from cascaded smoothing can be different per class (Q7 also discusses a related point). A possible reason why such a variability exists is that the model itself, i.e., either the denoiser or classifier, is biased towards certain features of a class, and this may affect the performance of denoised smoothing if the features were less generalizable, e.g., from backgrounds. Our proposed denoiser fine-tuning is an attempt to address this point particularly focusing on the denoiser part: specifically, we propose anti-consistency loss to penalize when a denoiser confidently outputs a different class from the input - Figure 1(b) in our manuscript illustrates an example, for your information.
>
> ---
>
> **Q6. Consistency of cascading over noise samples**
>
> We implement our proposed cascaded smoothing to make “statistically consistent” predictions across different noise samples, considering a certain _significance level_ $\alpha$ (e.g., $\alpha = 0.001$ in our experiments) - in a similar fashion as Cohen et al. [1]. Roughly speaking, for a given input $\mathbf{x}$, it makes predictions only when the confidence interval of $p_{\hat{f}}(\mathbf{x})$ does not overlap with $p_0$ upon $n$ _i.i.d._ noise samples and $\alpha$ (otherwise it abstains). The more details on the certification procedure can be found in Appendix C.2 for your information, and we will further highlight this in the final draft, e.g., by also providing a pseudocode.
>
> ---
>
> **Q7. “Critical” $\sigma$, and its distribution**
>
> Although it is not required for our cascaded smoothing itself to ensure the stability you mentioned across all $\sigma$, e.g., outside $\\{0.25, 0.5, 1.0\\}$, we indeed observe that smoothed confidences observed from denoised smoothing usually interpolate smoothly between different $\sigma$’s, at least for the diffusion denoised smoothing pipeline - in other words, one can also consider a concept of “critical” $\sigma$ as you mentioned, by measuring the threshold of $\sigma$ where its confidence goes below, e.g., 0.5. Figure S1 in the rebuttal PDF examines this measure on CIFAR-10 training samples and plots its distribution for each class as histograms: interestingly, we observe that the distributions are often significantly biased, e.g., the “Ship” class among CIFAR-10 classes relatively obtains much higher critical $\sigma$ - possibly due to its peculiar background of, e.g., ocean. We appreciate your insightful question, and we will incorporate this result and its respective discussion in the final draft.
>
> ---
>
> **References**
>
> [1] Cohen et al., Certified adversarial robustness via randomized smoothing, ICML 2019.

---

> > ### Comment · Reviewer_Vnny · 2023-08-17
> >
> > Thank you for responding to my comments and questions. While the answers to Q1 and Q2 and Q7 are convincing, the arguments on Q3, Q4 and Q5 do not address the fundamental issues. Therefore, I keep my initial score of borderline accept.

---

> > > ### Author Response · Authors · 2023-08-18
> > > **Thank you for the feedback**
> > >
> > > Dear Reviewer Vnny,
> > >
> > > We appreciate your response, and your acknowledgment of our answers. Regarding Q3, Q4, and Q5, could you please provide more detailed information on the fundamental issues, if possible? Your insights are invaluable, and having further clarity on them will greatly help us to strengthen our manuscript.
> > >
> > > Thank you very much for your time and effort again on our paper!
> > >
> > > Authors

---

### Author Rebuttal · Authors · 2023-08-09

Dear reviewers,

We thank all the reviewers’ efforts to improve our manuscript. This common response addresses a concern raised by multiple reviewers, viz., on our diffusion fine-tuning scheme compared to classifier fine-tuning. We also kindly ask you to find the attached rebuttal PDF in this response, which contains figures and tables to support our responses.

---

**[Vnny, b42a, K2e2] Fine-tuning denoiser vs. classifier**

Despite its effectiveness to improve denoised smoothing, as previously verified by Carlini et al. [1], we would like to highlight that classifier fine-tuning itself can be less practical when it comes with the cascaded smoothing pipeline proposed in this paper (in particular, the issue becomes more severe when dealing with larger classifiers):

- (a) To perform cascaded smoothing at multiple noise scales, classifier fine-tuning would require separate runs of training for optimal models per scale, also resulting in multiple different models to load - which can be less scalable in terms of memory efficiency.
- (b) In a wider context of denoised smoothing, the classifier part is often assumed to be a model at scale, even covering cases when it is a “black-box” model, e.g., public APIs. Classifier fine-tuning, in such cases, can become prohibitive or even impossible.

With respect to (a) and (b), the denoiser fine-tuning we propose offers a more efficient inference architecture: it can handle multiple noise scales jointly with a single diffusion model, while also being applicable to the extreme scenario when the classifier model is black-box. In other words, we argue that our proposed denoiser fine-tuning can be a scalable alternative to the classifier fine-tuning in practical scenarios.

Nevertheless, if one wishes, fine-tuning both classifiers and denoiser can bring their complementary effects to improve denoised smoothing: e.g., as we further examined in Table S1 of the rebuttal PDF, we observe that additionally fine-tuning classifiers (viz., to tune the classifier on denoised inputs rather than clean ones upon the diffusion fine-tuning) could further improve the overall ACR of the classifier. We also observe that in our experiments, denoiser fine-tuning itself could obtain a comparable gain in ACR to classifier fine-tuning, while the former can be favorable over the latter to use in practice due to (a) and (b).

In essence, the (somewhat counter-intuitive) effectiveness of denoiser fine-tuning we verify confirms that denoising process (e.g., via diffusion models) can be also biased as well as classifiers to make over-confident errors, where our proposed anti-consistency loss could play a role, e.g., as in Table S2 of the rebuttal PDF. We believe this observation can not only advance the performance, but also opens up a new direction of de-biasing diffusion models through the lens of randomized smoothing.

In the final draft, we will clarify these points and incorporate the additional results with the respective discussions.

---

Thank you for your consideration,

Authors

---

**References**

[1] Carlini et al., (Certified!!) Adversarial Robustness for Free!, ICLR 2023.

---

### Decision · Program_Chairs · 2023-09-21

**Decision:**

Accept (poster)

**Comment:**

The AC has carefully read the paper, reviews, author response, and the discussions. This work aims to improve the robustness-accuracy tradeoff in image classifiers by proposing two techniques for improving adversarial robustness without loss of accuracy. Their method has been evaluated on CIFAR10 and ImageNet datasets. All reviewers are positive towards this paper, including the clarity of writing, novelty, experimental results, and the importance of the problem. The AC agrees with the reviewers that the current form of the paper passes the bar of NeurIPS and thus recommends acceptance. However, the authors are strongly encouraged to follow reviews to polish the paper, for example, adding more comparisons between diffusion fine-tuning scheme and classifier fine-tuning.